

# Added value of geophysics-based soil mapping in agro-ecosystem simulations

Cosimo Brogi[1], Johan A. Huisman[1], Lutz Weihermüller[1], Michael Herbst[1], Harry Vereecken[1]

[1]Agrosphere (IBG-3), Institute of Bio- and Geosciences, Forschungszentrum Jülich, Germany

5    *Correspondence to*: Cosimo Brogi (c.brogi@fz-juelich.de)

**Abstract.** Developments in agricultural applications have led to an increased demand for quantitative high-resolution soil maps that enable within-field management. Commonly available soil maps are generally not suited for this purpose, but digital soil mapping and geophysical methods in particular allow to obtain soil information with unprecedented level of detail. However, it is often difficult to quantify the added value of such high-resolution soil information for agricultural 10 management and crop modelling. In this study, a detailed geophysics-based soil map was compared to two commonly available general-purpose soil maps. In particular, the three maps were used as input for crop growth models to simulate leaf area index (LAI) of five crops for an area of ~1 km2. The simulated development of LAI for the five crops was evaluated using LAI obtained from multispectral satellite images. Overall, it was found that the geophysics-based soil map provided better LAI predictions than the two general-purpose soil maps in terms of correlation coefficient R2, model efficiency (ME), 15 and root mean square error (RMSE). Improved performance was most apparent in case of prolonged periods of drought and was strongly related to the combination of soil characteristics and crop type.

## 1 Introduction

Detailed soil information on areas within a single field that require different treatment, so called management zones, is key in agricultural management (King et al., 2005;Stafford et al., 1996;Sylvester-Bradley et al., 1999). An adequate 20 characterization of such management zones can improve agricultural productivity and sustainability (King et al., 2005;Sylvester-Bradley et al., 1999) and help meeting future food security challenges (Antle et al., 2017;Chartzoulakis and Bertaki, 2015). Technological advances in geo-referencing, sensing, and computing have led to an increased possibility to apply distinct within-field management strategies (Sylvester-Bradley et al., 1999). As a consequence, there is a growing demand for high-resolution soil maps that can substitute national and regional scale soil maps (Brevik et al., 2006), which 25 are sometimes not available and often not suitable for agricultural applications and modelling (Della Chiesa et al., 2019;Nussbaum et al., 2018;Pätzold et al., 2008). However, high-resolution soil maps are generally costly to produce and the added value of such detailed mapping products is typically difficult to assess.



Management zones are generally characterized by soils with relatively uniform characteristics (i.e. a soil unit). These soil units can potentially be obtained from a variety of commonly available thematic maps that provide spatially distributed soil information (e.g., geological, soil, and yield potential maps). However, these products are often insufficiently detailed (Franzen et al., 2002;Nawar et al., 2017;Robert, 1993) since they are discretized in relatively large polygons and provide qualitative information that might differ from the inputs that are useful for farmers (Krüger et al., 2013;Söderström et al., 2016). This is generally a consequence of the sparse point-scale soil sampling on which most of these maps are based (Gebbers and Adamchuk, 2010;Heuvelink and Webster, 2001), which is ~1 point per ha for highly detailed products (Rogge et al., 2018). Maps with higher sampling resolution such as the German soil taxation maps that were surveyed on a 50 m grid (four points per ha) often do not provide improved results (Mertens et al., 2008). Denser sampling can be locally applied in order to obtain a more detailed and reliable soil characterization. However, this is time- and resource-consuming (King et al., 2005), and it is desirable to find more cost-effective mapping tools (Brevik et al., 2006).

Geophysical methods such as electromagnetic induction (EMI) have proven their potential in assisting agricultural applications (Binley et al., 2015) by providing a suitable alternative to dense soil sampling (Robinson et al., 2008). EMI measures the apparent electrical conductivity of the soil (ECa), which can be used to estimate soil characteristics and properties such as water content, textural properties, mineralization, porosity, and residual pore water content (Corwin and Lesch, 2005). Due to its high mobility, EMI can provide maps that range from the field to the catchment-scale (Robinson et al., 2008). Despite these promising aspects, EMI has often been credited primarily for qualitative mapping (Binley et al., 2015). However, there is renewed interest in this technique because of the development of multi-coil instruments that allow to investigate multiple depths simultaneously (Monteiro Santos et al., 2010;Saey et al., 2012;von Hebel et al., 2014). Furthermore, EMI has already shown high potential for the determination of management zones (Brogi et al., 2019;Galambošová et al., 2014;King et al., 2005;Moral et al., 2010;Oldoni and Bassoi, 2016;Taylor et al., 2003;Terrón et al., 2015). Additional information about soil horizonation and texture can be added to EMI-derived maps using direct soil sampling and laboratory analysis (Brogi et al., 2019).

One drawback of EMI-derived soil maps is that they can only determine potential management zones without directly providing information on the appropriate management of such areas (King et al., 2005). To increase their usefulness, geophysics-based soil maps can be used as input for process-oriented crop growth models (Brogi et al., 2020;Krüger et al., 2013) that account for factors limiting crop growth, such as water availability (Bonfante et al., 2015;Paz, 2000). Most process-oriented crop growth models rely on a one-dimensional description of water flow in the soil column (Vereecken et al., 2016) and require detailed information on soil profile characteristics, including soil hydraulic properties (Boenecke et al., 2018). Some case studies where EMI-based soil maps were successfully combined with crop growth models are available (Boenecke et al., 2018;Wong and Asseng, 2006). Recently, Brogi et al. (2020) successfully used inputs from an EMI-derived soil map to simulate soil water content dynamics and their effects on the growth of six crop types for a 90 ha study area.

Furthermore, Krüger et al. (2013) showed that the consideration of soil depth derived from geophysical measurements improved the simulation of biomass production on a 4.4 ha experimental field compared to simulations based only on commonly available soil maps. Despite these promising results, only a few studies linked geophysics-based soil maps and crop growth models. Moreover, there is a general lack of research aiming at the quantification of the added value of geophysics-based soil characterization for crop modelling applications (Krüger et al., 2013), especially in areas larger than the field-scale and for multiple crop and soil types.

The aim of this study is to assess the added value of a detailed soil map for agricultural applications in comparison to the use of commonly available soil maps. For this, a recently produced geophysics-based soil map (Brogi et al., 2019), a 1:5000 regional soil map (Röhrig, 1996), and a national soil taxation map (NRW, 1960) were used as input for the agro-ecosystem model AgroC. Simulations were made for five crops (i.e. silage maize, sugar beet, winter barley, winter rapeseed, and winter wheat) grown in 2016 on an area of approximately 1 x 1 km where water scarcity is known to have an effect on crop development (Rudolph et al., 2015). In a first step, the information provided by the three soil maps was compared. Next, AgroC simulations of leaf area index (LAI) based on the three soil maps were evaluated using LAI observations derived from multispectral satellite data. Finally, the added value of the geophysics-based soil map for the simulation of crop growth in the study area was investigated and discussed with a focus on the results for sugar beet.

## 2 Materials and methods

### 2.1 Study area

The study site is located within a ~1 x 1 km area (Fig. 1a) in the Rur catchment near Selhausen (Germany). It is composed of several agricultural fields cultivated in rotation by more than 20 different land owners, and thus the agricultural management is rather heterogeneous. The mean annual temperature and precipitation are 10.2°C and 715 mm, respectively (Rudolph et al., 2015). The area is part of the Terrestrial Environmental Observatories (TERENO) network (Bogena et al., 2018;Schmidt et al., 2012;Simmer et al., 2015). Within this network, continuous measurements of meteorological parameters are performed in the center of field F11 (Fig. 1a).

In 2016, the 44 investigated fields were cropped with silage maize, sugar beet, winter barley, winter rapeseed, and winter wheat. The distribution of these crops in the study area is shown in Fig. 1b. The total area cultivated with each crop as well as their approximate emergence and harvest date are listed in Table 1. A more detailed investigation was performed in fields where sugar beet was grown (fields F01, F05, F13a, F46, F48, F49, F12, F47, F50, and F51 in Fig.1 a).

Previous studies at this site showed that soil heterogeneity has a strong influence on crop performance during long periods of water scarcity (Rudolph et al., 2015;von Hebel et al., 2018). This is most apparent on the upper terrace, which has an altitude



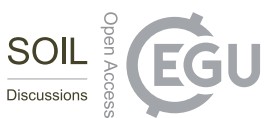

of ~110-113 m a.s.l. and is separated from the lower terrace (~101-103 m a.s.l) by a gentle slope with western exposition
(Fig. 1). For both terraces, aeolian Pleistocene sediments and translocated Holocene loess (Röhrig, 1996) are generally
dominant in the top soil. Within the top 2 m of soil, sand and gravels from the Pleistocene and Holocene are consistently
found below the aeolian sediments of the upper terrace and locally below the top soils of the lower terrace (Brogi et al.,
2019;Klostermann, 1992;Pätzold et al., 2008;Röhrig, 1996). The depth to these sand and gravels is known to be a strong

control on crop water availability (Brogi et al., 2020;Rudolph et al., 2015;von Hebel et al., 2018).

Information on the spatial and temporal development of the five crops in 2016 was derived from multispectral satellite
images. In particular, six LAI maps estimated from Level 3A RapidEye data were available. These images covered the
growing season of the five investigated crops and were acquired on the 14th of March, 20th of April, 28th of May, 9th of June,

12th of August, and 8th of September 2016. In order to generate the LAI maps, the normalized difference vegetation index
(NDVI) of the satellite images was first calculated using the red (RED, 630 – 685 nm) and the near infrared (NIR, 760 – 850
nm) bands. In a second step, NDVI values were used to calculate the fractional vegetation cover ($FVC_{NDVI}$) for each image
using the NDVI of bare soil ($NDVI_S$) and the fully vegetated state ($NDVI_V$) (Beck et al., 2006;Xiao and Moody, 2005;Zeng
et al., 2003). Finally, the $LAI_{NDVI}$ was calculated from the $FVC_{NDVI}$. In this final stage, the light extinction coefficient $k(\theta)$ of

each crop type, which is a measure of attenuation of radiation in the canopy (Campbell, 1986;Norman and Campbell,
1989;Ross, 2012;Propastin and Erasmi, 2010), was calibrated using 45 *in situ* destructive LAI measurements that were
acquired between the 22nd of March and the 7th of September 2016. In the case of winter rapeseed and silage maize, no *in situ*
LAI measurements were available and $k(\theta)$ was obtained from literature values (Ali et al., 2015). A detailed description of
the procedure used for this study area can be found in Brogi et al. (2020), whereas the general approach is described in more

detail in Ali et al. (2015).

**2.2 Available soil maps**

Three different soil maps of the same area were used to inform agro-ecosystem simulations. Simulations were first
performed using information from a geophysics-based soil map (Brogi et al., 2019) employing a methodology that was
already successfully applied in this area by Brogi et al. (2020). In a following step, simulations were performed using

information derived from two commonly available soil maps: i) a regional soil map (Röhrig, 1996) with a scale of 1:5000
and ii) a national soil and yield potential map (NRW, 1960) used in agricultural taxation.

The geophysics-based soil map (Fig. 2a) is a high-resolution product that was obtained by combining multi-configuration
EMI measurements and direct soil sampling with subsequent laboratory analysis (Brogi et al., 2019). EMI measurements

were performed using a CMD Mini Explorer - Special Edition (GF Instruments, Brno, Czech Republic) and resulted in six
ECa maps, each with a different depth of investigation. These maps were analyzed using a supervised classification
methodology. This resulted in a map that showed the spatial distribution of 18 different soil types within the study area.





Quantitative soil information was obtained using 100 augering locations (~one location per ha) where horizon type and thickness were recorded. At these locations, soil samples were also collected to determine the grain size distribution of each

soil horizon. Finally, information obtained through direct soil sampling was combined with the spatial distribution of the ECa-based soil units, which resulted in a soil map with 1 m resolution and quantitative soil information up to 2 m depth. As shown in Fig. 2a for the soil units A1a and C1a, each soil unit of the geophysics-based soil map is provided with a soil profile that shows the depth of each horizon as well as the textural information (percentage of clay, silt, and sand plus gravels) up to a depth of 2 m. This geophysics-based product was recently used in several studies focused on the Selhausen

study area (Brogi et al., 2020;Jakobi et al., 2020;Reichenau et al., 2020). The geophysics-based soil map consists of four sub-areas (A, B, C, and D) and a total of 18 soil units with an area between 0.6 and 16.0 ha.

The 1:5000 regional soil map is shown in Fig. 2b. This thematic map was first produced in 1984/85 and then revised in 1996 (Röhrig, 1996). It is broadly used in regional projects (e.g., sustainable soil protection plans) and is part of the NRW official

soil inventory (NRW, 2018). The topographic base is the 1:5000 German Land Map (Deutsche Grundkarte) and soil information is obtained through direct soil sampling (augering). The distance between auger locations is typically <100 m (~one location per ha). For each soil unit, information on soil type, soil texture (qualitative description), and thickness of the top soil horizons is provided. Generally, the depth of the interface between two soil horizons is represented by a range (e.g., in soil unit A-1B of Fig. 2b this interface is found between 0.3 m and 0.6 m depth). This information was generalized by

averaging the reported maximum and minimum values. According to this map, 13 soil units are found in the study area with sizes between 1.0 and 25.3 ha.

The soil taxation map is shown in Fig. 2c. It is used for the calculation of tax rates for farmers and land owners based on estimates of the yield potential (NRW, 1960). The map is based on the German Cadaster and soil information was obtained

through direct soil sampling with a separation of 40 to 50 m between augering locations (~four locations per ha). Each soil unit is described with a 2.0 m depth soil profile, which is divided in up to four horizons. Each horizon carries qualitative information on soil texture. In some cases, the interface between two horizons is represented by a range which again was generalized by averaging the reported maximum and minimum values (e.g. soil unit A-01 in Fig. 2c). In total, 10 different soil units were identified in the study area and their area ranges between 0.7 and 17.5 ha.


The area covered by the five investigated crops in 2016 (Fig. 1b) and the geometry of the three soil maps (Fig. 3) were intersected. As a result, a set of unique soil-crop combinations was obtained for each soil map: i) 72 unique soil-crop combinations for the geophysics-based soil map, ii) 42 for the 1:5000 soil map, and iii) 35 for the soil taxation map.





## 2.3 Similarities between sub-areas of the three soil maps

According to the geophysics-based soil map, the study area is divided into four sub-areas (A, B, C, and D). Sub-area A is located on the upper terrace, sub-area B is located on the slope between the two terraces, and sub-areas C and D are located on the lower terrace. Crop growth simulations of sub-areas B and C by Brogi et al. (2020) showed similar results. Thus, they were described together as a single sub-area BC in this study. Generally, the geometry of the soil units described in the commonly available soil maps falls within one of the four sub-areas of the geophysics-based product. Thus, the soil units of

the commonly available maps were matched with one of the sub-areas A, BC, and D. As a result, a clearer comparison between the simulations and the results obtained with different maps was possible. The distribution of the soil units described in the three soil maps is shown in Fig. 3 and the final unified code of each soil unit is provided in Table 2.

## 2.4 The AgroC model

In this study, the AgroC model (Herbst et al., 2008) was used to simulate crop growth within each of the soil-crop
combinations. In particular, we were interested in investigating how differences in soil water content dynamics and water availability as a consequence of differences in soil layering and soil hydraulic parameters affected crop growth. The agro-ecosystem model AgroC couples three main modules: SOILCO2, SUCROS, and RothC. SOILCO2 simulates vertical fluxes of water, heat, and $CO_2$ for a 1D soil column (Šimůnek et al., 1996; Šimůnek and Suarez, 1993), SUCROS simulates crop growth (Spitters et al., 1989), and RothC simulates organic carbon turnover (Coleman and Jenkinson, 1996).


In the SOILCO2 module, water flow in a given soil profile is described by the 1-dimensional Richards equation:

$$\frac{\partial \theta}{\partial t} = \frac{\partial}{\partial z}\left[K(h)\left(\frac{\partial h}{\partial z} - 1\right)\right] - Q \tag{1}$$

where $\theta$ (cm³ cm⁻³) is the volumetric water content, $t$ is time, $z$ is the vertical coordinate (cm), $K(h)$ (cm h⁻¹) is the hydraulic conductivity as a function of pressure head $h$ (cm), and $Q$ (cm³ cm⁻³ h⁻¹) is the source/sink term accounting for root water
uptake by crops. The hydraulic conductivity $K$ and the volumetric water content $\theta$ as a function of pressure head are described by the Mualem- van Genuchten model:

$$K(h) = K_s S_e^{1/2}\left[1 - \left(1 - S_e^{1/m}\right)^m\right]^2 \tag{2}$$

and

$$\theta(h) = \theta_r + \frac{\theta_s - \theta_r}{(1 + |\alpha h|^n)^m} \tag{3}$$

where $K_s$ is the saturated hydraulic conductivity (cm h⁻¹), $m$ is a parameter that is set equal to 1-1/$n$, $\theta_s$ and $\theta_r$ (cm³ cm⁻³) are the saturated and residual water content, respectively, $\alpha$ is the inverse of the air entry pressure (cm⁻¹), $n$ (dimensionless) is a



parameter related to the pore size distribution (Van Genuchten, 1980), and $S_e$ is the dimensionless relative saturation defined as:

$$S_e = \frac{\theta - \theta_r}{\theta_s - \theta_r} \tag{4}$$

Root water uptake by crops is considered through the sink term $Q$ in Eq. (1). The water demand is calculated in the SUCROS module. For a crop growing under optimal conditions, the potential evapotranspiration is divided between potential transpiration $T_p$ (cm h$^{-1}$) and potential soil evaporation $E_p$ (cm h$^{-1}$). Then, the potential root water uptake $S_p$ (cm$^3$ cm$^{-3}$ h$^{-1}$) is calculated using:

$$S_p(z) = \text{ß}(z) T_p \tag{5}$$

where ß is the depth-dependent root distribution function. Afterwards, the depth-specific actual root water uptake $Q$ is obtained from the potential root water uptake $S_p$ using:

$$Q(z, h) = \varphi(h) S_p(z) \tag{6}$$

where the water uptake is scaled by a water uptake stress factor $\varphi$ in dependence of pressure head. The $\varphi(h)$ factor is calculated using the Feddes et al. (1978) approach:

$$\varphi(h) \begin{cases} \frac{h_0 - h}{h_0 - h_1} & h_0 \leq h \leq h_1 \\ 1 & \text{for } h_1 \leq h \leq h_2 \\ 10^{\frac{h_2 - h}{h_3}} & h_2 \leq h \leq h_3 \end{cases} \quad \text{being} \begin{array}{l} h_0 = 0 \\ h_1 = -20 \\ h_2 = -5000 \\ h_3 = -16000 \end{array} \tag{7}$$

where $h_{0-3}$ (cm) are threshold pressure heads obtained from literature (Vanclooster et al., 1995). The actual root water uptake is integrated over the rooting depth to obtain the actual transpiration. Afterwards, the ratio between actual and potential transpiration is used to simulate carbon assimilation and biomass production, which are also affected by additional variables such as temperature and solar radiation. A detailed description of this model and of the effects of simulated water stress on 205 simulated crop growth can be found in Klosterhalfen et al. (2017) and in Brogi et al. (2020) for this study area.

**2.5 Estimation of soil hydraulic parameters**

Soil hydraulic parameters for each horizon of the soil units of all three soil maps were estimated from texture and bulk density using a pedotransfer function (PTF) and a correction for gravel content as successfully done by Brogi et al. (2020) for the same area. Information on bulk density was not available. Therefore, the bulk density of the fine fraction < 2 mm, 210 BD$_{<2}$, of each horizon was assumed to be 1.30 g cm$^{-3}$ for the Ap horizon, 1.40 g cm$^{-3}$ for the AB horizon, 1.50 g cm$^{-3}$ for deeper horizons with fine sediments, and 1.60 g cm$^{-3}$ for deeper horizons with coarse sediments. These estimates were based on literature values and on results from previous sampling campaigns conducted within the study area (Brogi et al., 2020;Ehlers et al., 1983;Unger and Jones, 1998). Unfortunately, the soil profiles from the 1:5000 soil map and from the soil



taxation map are not provided with the depth of the interface between the Ap and AB. Therefore, in each soil profile, the first
horizon was subdivided at a depth of 0.3 m as this was generally observed to be the depth of the Ap horizon in this study
area.

The geophysics-based soil map provides quantitative information on soil texture for each soil unit. In this case, the
percentage of clay, silt, and sand were directly used in the estimation of the soil hydraulic parameters. On the contrary, the
two commonly available soil maps provide a qualitative description of soil texture (i.e. a soil textural class). Generally, look-
up tables are used in case of such qualitative descriptions (Van Looy et al., 2017) in order to obtain class-average soil
hydraulic parameters (Baker, 1978;Bouma, 1989). To obtain a consistent comparison between the three maps, these
qualitative soil textural class descriptions were translated in quantitative percentages of sand, silt, and clay by using the
USDA soil textural classification (USDA, 2019). For this, the centroid of each soil textural class within the USDA triangle
was calculated to determine the associated soil texture percentages (Table 3). In some cases, the presence of gravel was
qualitatively described in the two commonly available soil maps. In these cases, a 25% volume of gravel was assumed when
the gravel content was defined as "gravelly" and 10% was assumed when the gravel content was defined as "weakly
gravelly" as these percentages matched those observed in the study area.

The dry bulk density was estimated for each soil horizon using the equation of Brakensiek and Rawls (1994):

$$BD_t = BD_{<2} + G_v (BD_{>2} − BD_{<2}) \tag{8}$$

where $BD_t$ is the dry bulk density of the soil, $BD_{>2}$ is the dry bulk density of gravel material, and $G_v$ is the volume of gravel
calculated from the percentage of weight according to Flint and Childs (1984). Finally, the soil hydraulic parameters $\theta_s$, $\theta_r$, $\alpha$,
$n$, and $K_s$ were estimated from the dry bulk density and from clay, silt, and sand percentages by using the pedotransfer
function of Rawls and Brakensiek (1985). In all the soil maps, the estimated $K_s$ of the coarse horizons in sub-areas A and D
were corrected for gravel content using the correction from Brakensiek and Rawls (1994):

$$K_b = K_s[2(1 − G_v )/(2 + G_v )] \tag{9}$$

were $K_b$ is the saturated hydraulic conductivity of the bulk soil and $K_s$ is obtained from the PTF of Rawls and Brakensiek
(1985).


As shown in Brogi et al. (2020), the coarse sand and gravel horizon 2C that underlies the fine aeolian sediments in sub-area
A and in parts of sub-area D, is of primary importance for a sound simulation of crop performance within the investigated
area. However, the properties of these deeper soil horizons were not adequately captured in the commonly available soil
maps, and this resulted in simulated water contents that were unrealistically low. To avoid the introduction of such strong
and unrealistic variations in the results of the agro-ecosystem simulations performed with the three soil maps, the soil



hydraulic parameters obtained for the 2C horizon in sub-area A and D of the geophysics-based soil map were integrated into the two other commonly available soil maps.

## 2.6 Setup and evaluation of AgroC simulations

The 1-dimensional soil column used in the AgroC simulations was discretized by using a maximum of 252 nodes with a
spacing of 1 mm near the soil surface and a gradual increase with depth until a maximum of 10 mm. The simulation period extended from 1$^{st}$ of July 2015 to 31$^{st}$ of December 2016. The simulation domain of the soil units in sub-area BC extended to 2.0 m below the soil surface for all three soil maps. In these units, a variable pressure head with an annual sinusoidal variation based on water table depth observations in field F10 (Fig. 1) was used as the lower boundary condition. The groundwater table depth was set to a minimum of -2.0 m on the15$^{th}$ of January and a maximum of -2.6 m on the 15$^{th}$ of July.
The initial pressure head within the soil profile was defined through a spin-up simulation. To achieve this, the period from 1$^{st}$ of January 2015 to 31$^{st}$ of December 2016 was repeatedly run until no change in pressure head within the soil column was observed between consecutive model runs.

The soil units of sub-areas A and D have a coarse sand and gravel horizon at depth. In these soil units, the simulation domain
extended 30 mm into the coarse horizon. As a result, the simulation domain varied in depth between 0.52 and 1.57 m for the geophysics-based soil map and from 0.33 to 1.43 m in the two commonly available soil maps. Free drainage was used as the lower boundary condition for these soil units. The initial pressure head of the lower coarse horizons was set to -10 mm and hydrostatic equilibrium was assumed throughout the profile (Brogi et al., 2020).

The required crop-specific parameters were mostly obtained from various literature sources (Allen et al., 1998;Bolinder et al., 1997;Boons-Prins et al., 1993;Borg and Grimes, 1986;Penning de Vries et al., 1989;Spitters et al., 1989;Van Heemst, 1988;Vanclooster et al., 1995). Only the sub-area specific parameterization of the death rate of leafs for sugar beet, the start of the senescence stage for silage maize, and the partitioning of aboveground biomass for winter wheat were adapted as described in detail in Brogi et al. (2020).

The crop-specific maximum rooting depth was set to 1.50 m for sugar beet and silage maize, 1.40 m for winter rapeseed, 1.20 m for winter barley, and 1.00 m for winter wheat. The method of Rum et al. (1974) was used to calculate the root distribution above these depths. In the case of the soil units of sub-area A and D, rooting depth was reduced using the depth of the lower coarse horizon as it was assumed that roots cannot penetrate into such coarse soils (Daddow and Warrington,
275  1983).

Three criteria were used to quantify the agreement between the observed LAI$_{NDVI}$ and LAI simulated with AgroC by using inputs from the three soil maps. The first was the root mean square error (RMSE):





$$RMSE = \sqrt{\frac{1}{n}\sum_{i=1}^{n}(Obs_i - Sim_i)^2} \qquad (10)$$

A lower RMSE indicates a better fit. The second criteria was the model efficiency (ME), which is calculated as:

$$ME = \frac{\sum_{i=1}^{n}(Obs_i - \overline{Obs})^2 - \sum_{i=1}^{n}(Obs_i - Sim_i)^2}{\sum_{i=1}^{n}(Obs_i - \overline{Obs})^2} \qquad (11)$$

where $\overline{Obs}$ is the observed mean. The ME can vary between $-\infty$ and 1 and a value higher than 0 indicates that the model describes the data better than the mean. A ME value of 1 indicates perfect agreement between observations and predictions (Nash and Sutcliffe, 1970). The third and final criteria was the coefficient of determination $R^2$:

$$R^2 = [\frac{\sum_{i=1}^{n}(Obs_i - \overline{Obs})(Sim_i - \overline{Sim})}{\sqrt{\sum_{i=1}^{n}(Obs_i - \overline{Obs})^2 \sum_{i=1}^{n}(Sim_i - \overline{Sim})^2}}]^2 \qquad (12)$$

where $\overline{Sim}$ is the simulated mean. The value of $R^2$ ranges between 0 and 1, and a value of 1 indicates perfect agreement.

## 3 Results and discussion

### 3.1 Comparison of the spatial distribution of soil properties in the three maps

By visually comparing the three soil maps, it is apparent that the geomorphological border between the upper and lower
terrace is similarly identified (division between sub-area A and BC in Fig. 3). In the geophysics-based soil map, this subdivision is based on the measured apparent electrical conductivity (ECa) data (Brogi et al., 2019), whereas the delineation in the commonly available soil maps was mainly based on coarse soil augering and topography since this limit approximately coincides with the top of the slope that divides these two sub-areas. The location of the subdivision between sub-areas BC and D showed stronger differences between the three maps. Again, this border is obtained from ECa data in the
geophysics-based soil map. In this case, there is no topographic feature associated with this subdivision and, in the other two maps, only the information from augering was available to determine its position. This likely explains why the subdivision between the two sub-areas locally coincides with field boundaries in the 1:5000 soil map and in the soil taxation map.

The geophysics-based soil map generally identifies a larger number of soil units which features more complex polygon
shapes compared to the commonly available soil maps. This is as a consequence of the high resolution of the measured ECa data. In this study area, a single agricultural field contained 4 to 9 soil units according to the geophysics-based soil map. The two commonly available soil maps often integrate larger areas into one soil unit and a single field is composed of a maximum of four soil units. However, many fields contain only one soil unit.

In the 1:5000 soil map and in the soil taxation map, the shallow soils found within the study area are generally silty loam and loamy silt above coarse sediments. As shown in Fig. 4, the grain size distribution obtained from the qualitative descriptions





using Table 3 showed minor differences compared to the one provided by the soil units of the geophysics-based soil. For example, the estimated grain size distribution for loamy silt soil units of the commonly available maps was 25% sand, 60% silt, and 15% clay, whereas the respective range of values found in the geophysics-based soil map was 13-24% sand, ~56-70% silt, and ~13-23% clay. The estimated vales for silty loam (22% sand, 70% silt, and 8% clay) differed more from those of the geophysics-based soil map, but still were in reasonable agreement. The underlying coarse materials found in sub-areas A and D were identified as loamy sand, sand, sandy clay loam, or sandy loam in the commonly available soil maps. In the case of these four soil textural classes, the sand fraction reported in Table 3 (60-90%) was much higher than the values reported in the geophysics-based soil map (~28-58%). At the same time, the percentage of silt in the commonly available maps (5-10%) was much lower compared to those of the geophysics-based map (~30-54%). Figure 4 illustrates these textural differences in sand and silt percentages between the three soil maps at a depth of 1.50 m.

The description of the top soil in the three soil maps is rather similar. On the contrary, the geophysics-based soil map provides a different description of the underlying coarse horizons compared to the commonly available soil maps. It is important to note that the textural compositions found in the geophysics-based soil map were determined from laboratory analysis, whereas those provided by the other two maps where generated from field estimations (hand texturing). Due to these differences, soil hydraulic parameters of the coarse sediments vary strongly between the three maps, with the commonly available maps showing much higher values of $K_s$ and $\theta_s$. As already mentioned in the methods section, this resulted in unrealistic simulations with very low soil water content for the commonly available maps (not shown). For this reason, the soil hydraulic parameters of the coarse horizons of sub-areas A and D of the geophysics-based soil map were integrated in the simulations based on the two commonly available soil maps.

Differences between the three soil maps in terms of soil layer thickness and depth were also observed (Fig. 5), especially for the depth of the coarse sand and gravel that underlies the silty sediments in sub-areas A and D (Fig. 5). In all three maps, this depth was obtained from augering information and varied between 0.47 and 1.34 m in the geophysics-based soil map, between 0.30 and 1.50 m in the 1:5000 soil map, and between 0.30 and 1.40 m in the soil taxation map. Between the two commonly available soil maps, the depths of the coarse horizon in the 1:5000 regional soil map were closer to that of the geophysics-based soil map.

### 3.2 Performance of LAI simulations

Figure 6 shows a comparison between the mean observed $LAI_{NDVI}$ for each soil crop-combination and the LAI simulated with AgroC by using inputs from the three soil maps. The LAI simulations based on the geophysics-based soil map ($R^2$ = 0.925, ME = 0.919, and RMSE = 0.604) were better able to describe $LAI_{NDVI}$ compared to the results of the simulation based on the 1:5000 soil map ($R^2$ = 0.887, ME = 0.869, and RMSE = 0.718) and the soil taxation map ($R^2$ = 0.886, ME = 0.866, and RMSE = 0.719). However, the results for all three soil maps could be considered satisfactory as the improvement



340 provided by the more detailed geophysics-based soil map is relatively small. This similarity in model quality can be explained by the simultaneous use of all simulated crops (silage maize, sugar beet, winter barley, winter rapeseed, and winter wheat) and all six RapidEye images. Furthermore, this comparison was performed by using the mean value for each soil-crop combination and this reduced the influence of small-scale variability of $LAI_{NDVI}$. These aspects will be investigated in more detail in the next section.

### 3.3 Pixel-by-pixel comparison of LAI simulations

Figure 7 shows a pixel-by-pixel comparison of observed $LAI_{NDVI}$ and LAI simulated using inputs from the three soil maps. It is apparent how the highest pixel density (red) is close to the 1:1 line for the geophysics-based soil map, and more spread is observed for the two commonly available soil maps. Moreover, the error measures for the geophysics-based soil map ($R^2$ = 0.884, ME = 0.878, and RMSE = 0.689) show slightly better results than those of the 1:5000 soil map ($R^2$ = 0.858, ME = 0.847, and RMSE = 0.774) and considerably better results than those of the soil taxation map ($R^2$ = 0.741, ME = 0.675, and RMSE = 1.126). The values of $LAI_{NDVI}$ occasionally showed a rather strong variability within single soil-crop combinations. As a consequence, lower values of $R^2$ and ME and higher values of RMSE are found in the pixel-by-pixel analysis compared to the previously discussed one that made use of the average $LAI_{NDVI}$ for each soil-crop combination.

355 Table 4 shows the pixel-by-pixel performance of the three soil maps for each RapidEye image. The geophysics-based soil map generally showed higher ME, and higher $R^2$ and lower RMSE. However, it is apparent that the three soil maps performed similarly in March, May, and June, when winter crops were grown (i.e. winter barley, winter rapeseed, and winter wheat) with the 1:5000 soil map showing marginally higher ME and $R^2$ for individual dates. Given the sufficient amount of rain during the growth of these crops, simulations showed little crop water stress. This resulted in simulations with rather similar LAI despite using information from three different soil maps. In the case of the image from April, the geophysics-based soil map and the 1:5000 soil map outperformed the soil taxation map. However, a general decrease in performance can be observed for all three maps. This general decrease is likely caused by the high variability in $LAI_{NDVI}$ between different agricultural fields which is caused by the uncertainty in seeding and emergence dates of winter crops. Moreover, the flowering of winter rapeseed (yellow dots in Fig. 6) affected the estimated $LAI_{NDVI}$ (Brogi et al., 2020). This may have additionally reduced the performance for all three soil maps in April. A strong drop in performance for all three soil maps was observed in August and September. At this stage, the comparison is based solely on silage maize and sugar beet as the other crop types were already harvested. Thus, the reduced correlation is likely related to the appearance of stress in these two crops that increased the spatial variability in $LAI_{NDVI}$. In this situation of strong water stress, the geophysics-based soil map clearly outperformed the 1:5000 soil map and the soil taxation map, especially towards the end of the growing season.



## 3.4 Simulation of sugar beet

In a next step, simulations of sugar beet are analyzed in more detail because of the importance of this crop (sugar beet represents 31.7% of the investigated area) and because the previously described analysis showed that the difference in performance between the three soil maps was strongest when this crop was still growing (August and September).

Figure 8a-c shows the simulated LAI (lines) and the observed $LAI_{NDVI}$ (dots) for sugar beet grown on the soil units of sub-area A for the three soil maps. Figure 8d-f shows the associated water stress simulated with AgroC. All these simulations made use of the same crop parameterization. Thus, the differences in simulated LAI are due to different degrees of water stress associated with the soil characteristics. As shown in Fig. 8a, the simulated LAI based on the geophysics-based soil map matched well with the observations. In fact, soil units A1a-d showed very similar LAI values at the 28th of May and the 9th of June. Later in the growing season (12th of August and 8th of September), simulated LAI of soil units A1a-d differed due to a different magnitude of water stress. The magnitude of water stress and the consequent reduction of LAI are well correlated with the depth of the coarse sediments, which was 0.86 m in soil unit A1a, 0.67 m in A1b, 0.58 in A1c, and 0.49 m in A1d.

In the case of the three soil units of the 1:5000 soil map, the simulated LAI well matched the observations in May and June with the only exception of soil unit A-07L where observed $LAI_{NDVI}$ is underestimated. Later in the growing season (August and September), soil unit A-7L again matched the observations. This was not the case for soil units A-1B and A-2B where $LAI_{NDVI}$ was underestimated. The mismatch between LAI simulations and observations for these two soil units is due to the assumed shallow depth to the coarse sediments (0.30 m in soil unit A-2B, 0.45 m in unit A-1B, and 0.80 m in A-7L). As previously discussed, this depth has a strong influence on the intensity and timing of water stress. Generally, the top soil and the associated physical properties described in the 1:5000 soil map were rather similar to those of the geophysics-based soil map. As a consequence, the reduced match between simulated LAI and observed $LAI_{NDVI}$ was caused by an underestimation of the depth to the coarse sediments in soil units A-1B and A-2B which resulted in stronger water stress and lower LAI.

In the case of the soil taxation map, the observed $LAI_{NDVI}$ for the two soil units A-01 and A-03 were rather similar and showed a strong variability within each unit (see dots and error bars in Fig. 8c). This suggests that the geometry of the two soil units did not capture the spatial differences in crop performance present in the $LAI_{NDVI}$. The soil physical properties of the uppermost horizon in the soil taxation map were rather different compared to those of the geophysics-based soil map. In contrast, the soil physical properties of the coarse sediments were equal to those of the geophysics-based soil map by design. However, the depth to these coarse sediments in the two soil units was generally deeper (1.20 m in A-01 and 0.70 m in A-03) than in the geophysics-based soil map. Due to these different soil physical properties and the soil depth overestimation,



lower water stress in soil unit A-01 caused a higher simulated LAI in comparison to the observations. This depth was better captured in soil unit A-03 which showed better LAI simulations.

Figure 9 shows the spatial distribution of $LAI_{NDVI}$ for the 12th of August and the 8th of September. On both days, the large-scale pattern in $LAI_{NDVI}$ associated with the differences between sub-areas A, BC, and D were clearly visible. In August (Fig. 9a), the $LAI_{NDVI}$ reached the maximum observed value in sub-area BC, whereas lower values were observed in sub-area A due to water stress. In September (Fig. 9b), the development of sugar beet was affected by water stress in both sub-area A and D, as indicated by the lower $LAI_{NDVI}$ compared to sub-area BC.


The simulated LAI obtained using the three soil maps captured the large-scale pattern in $LAI_{NDVI}$ associated with the sub-areas A, BC, and D to some extent. The differences in crop performance between sub-area A and BC were sufficiently well represented in all three simulations. However, the border between low LAI in sub-area A and high LAI in sub-area BC is better represented in the geophysics-based approach (see field F47 in Fig. 9). The differences between high LAI of sub-area

BC and intermediate LAI of sub-area D were not well represented in the two commonly available soil maps. In fact, the simulated LAI is very similar in the two sub-areas when using the 1:5000 soil map and the soil taxation map. This is related to the generally low water stress simulated in sub-area D that did not result in a meaningful reduction of LAI. On the contrary, a substantial reduction of LAI can be observed in the simulations based on the geophysics-based soil map. The small-scale patterns in $LAI_{NDVI}$ observed within each sub-area were also partly captured by the geophysics-based soil map

for both dates. In contrast, simulations based on the other two soil maps did not provide result that could well represent these patterns. At the same time, it is clear that the small-scale variability in $LAI_{NDVI}$ is not fully captured in our simulation approach because variations within individual soil-crop combinations are not considered.

Table 5 provides the pixel-wise RMSE between simulated LAI and $LAI_{NDVI}$ for all 11 fields where sugar beet was grown.

Generally, these fields can be divided into three groups based on to the RMSE of each soil map. The first group consist of the fields F01, F05, F13, F46, F48, and F49. In these fields located in sub-area A, the geophysics-based soil map provided improved simulation results for both days compared to the simulations based on the two other soil maps as indicated by the lower RMSE. Field F46 was the only exception as the lowest values in August (RMSE = 0.52) was obtained with inputs from the 1:5000 soil map. In some cases, the difference in RMSE between the simulations based on the three soil maps was

rather limited such as in field F48 and field F49. Nevertheless, the RMSE obtained in this group of fields using inputs from the geophysics-based soil map was lower for both days and was larger in September than in August.

The second group consists of fields F12, F44, and F47 in sub-area BC. In these fields, no water stress was simulated for any of the soil maps. Therefore, all three soil maps provided accurate LAI simulations as indicated by the low RMSE. The third

group consist of fields F50 and F51 that are located in sub-area D. In August, the RMSE of the simulations performed using

the three soil maps was rather similar. On the contrary, the simulations based on the geophysics-based soil map clearly outperformed those based on the commonly available soil maps in September. This again corresponds with the simulated water stress in sub-area D, which was high in September but rather low in August.

Overall, the results clearly highlight that the improvements provided by the use of the geophysics-based soil map for the simulation of sugar beet are clearly dependent on the magnitude and timing of water stress. Simulations based on the three soil maps showed rather similar results in periods with enough water availability and in areas where water stress did not affect crop development. On the contrary, in the presence of significant water stress, the use of the geophysics-based soil map reduced the RMSE between LAI simulations and observations compared with the use of the 1:5000 soil map or the soil
taxation map. This reduction in RMSE varied substantially from field to field with a minimum of 2% and a maximum of 67%. Likely, this variability is due to local differences in the quality of the soil characterization provided by the three soil maps. The area-averaged RMSE reduction in periods of significant water stress was 25% compared to the 1:5000 soil map and 31% compared to the soil taxation map.

It has to be noted that the geophysics-based soil map was produced with a set of 100 sampling locations (one per ha), which is similar to the sampling density of the 1:5000 soil map (~one per ha) and lower than the one of the soil taxation map (~four per ha). It can be argued that the geophysics-based soil map had an advantage resulting from the quantitative textural information based on laboratory analysis. To a certain extent this holds true. However, it should be seen against the background that textural information on the coarse soils from the geophysics-based product was integrated into the
simulations with the commonly available soil maps. Furthermore, the characterization of the upper soils between the three maps only showed local differences.

The reduction in performance of the 1:5000 soil map and of the soil taxation map compared to the geophysics-based soil map was strongly dependent on local soil characteristics. In general, it was found that this reduction was caused by a poor
representation of i) the spatial geometry of individual soil units, ii) the depth to the coarse sediments, iii) the textural characteristics of the overlying sediments, and iv) the subdivision between large sub-areas. The different quality of this soil representation affected the quality of the LAI simulations throughout the year and especially towards the end of the growing season of the summer crops.

## 4 Conclusions

In this study, agro-ecosystem simulations were performed on a 1 x 1 km agricultural area by using information from three different soil maps: i) a high-resolution geophysics-based soil map, ii) a 1:5000 regional soil map, and iii) a soil taxation map. The three soil maps similarly subdivide the study area in three geomorphological sub-areas. However, substantial

differences were found in the identification of the boundary between these sub-areas. The top soil textural descriptions were rather similar in the three maps but the textural description of the underlying coarse horizons differed significantly. This coarse horizon is known to control crop water availability within the study area and unrealistic water content simulations were obtained when textural data from the commonly available soil maps were used for this coarse horizon. Therefore, the soil hydraulic parameters obtained from the geophysics-based soil map were integrated in the simulations based on the two commonly available soil maps for this horizon.

In a following step, the growth of silage maize, sugar beet, winter barley, winter rapeseed, and winter wheat was simulated using the agro-ecosystem model AgroC. The LAI simulated with this model was compared with LAI observations determined from RapidEye images using explained variance ($R^2$), the model efficiency (ME), and root mean square error (RMSE). The simulations based on the geophysics-based soil map consistently showed better results (higher $R^2$ and ME, lower RMSE). Thus, the use of the geophysics-based soil product outperformed the use of inputs from the two commonly available soil maps. However, simulation of winter crops and in periods with limited water stress showed subtle improvements only. In contrast, the geophysics-based soil map clearly outperformed the commonly available soil maps in periods with moderate to high water stress that caused a reduction in crop performance, particularly for silage maize and sugar beet.

AgroC simulations for sugar beet were analyzed in more detail. It was found that the commonly available soil representations were strongly outperformed by the geophysics-based product in areas with specific soils and in periods where stronger water stress was observed. This was related to a more accurate description in the geophysics-based soil map of: i) the depth to the coarse sediments, ii) the soil texture of the overlying horizons, iii) the subdivision of the four geomorphological sub-areas A, BC, and D, and iv) the distribution of the soil units that are found in the study area.

The geophysics-based soil characterization combined with direct soil sampling provided an added value to agro-ecosystem modelling and allowed for improved simulation of crop LAI and crop performance. However, simulations of prolonged periods of drought provided much more apparent improvements. At the same time, improvements were locally dependent on the combination of soil characteristics and crop type. Nevertheless, a substantial mismatch between the AgroC simulations and the satellite observations was observed when the two commonly available maps were used as input. Thus, a detailed and quantitative soil characterization such as a geophysics-based soil product has a positive and quantifiable utility in agriculture and, depending on local soil characteristics, is necessary for the use of advanced precision farming techniques and strategies that are not practicable when solely based on general-purpose soil maps.



## Acknowledgements

Support for this study was received through the "Terrestrial Environmental Observatories" (TERENO), the Advanced Remote Sensing—Ground-Truth Demo and Test Facilities (ACROSS) initiative, the Deutsche Forschungsgemeinschaft (DFG, German Research Foundation) under Germany's Excellence Strategy - EXC 2070–390732324 project PhenoRob, and the Deutsche Forschungsgemeinschaft through the Transregional Collaborative Research Center 32 – Patterns in Soil-Vegetation-Atmosphere Systems: Monitoring, Modeling and Data Assimilation.

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



**List of figures**

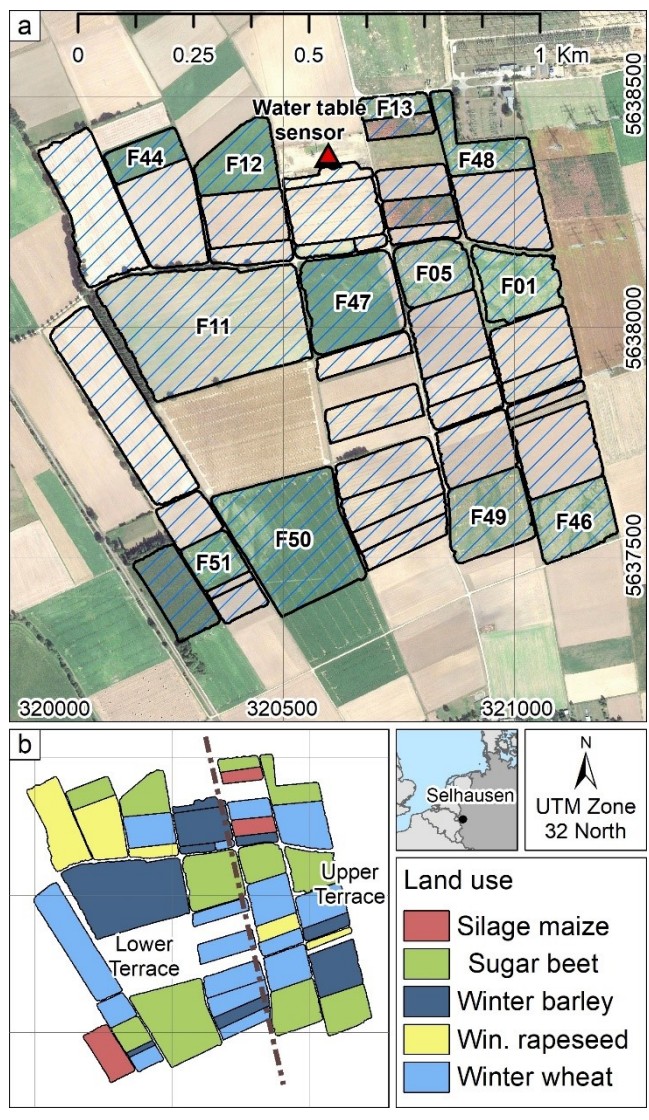

**Figure 1: a) satellite image of the study area with the fields used for comparison and codes of fields where sugar beet was grown in 2016, b) crops that were grown in each studied field in 2016 (ESRI, 2016).**







**Figure 2: a) geophysics-based soil map of the study area with examples of the quantitative description of soil units A1a and C1a, b) 1.5000 soil map of the study area with examples of the qualitative description of soil units A-1B and A-7L (Röhrig, 1996), and c) soil taxation map of the study area and examples of the qualitative description of soil units A-01 (NRW, 1960).**




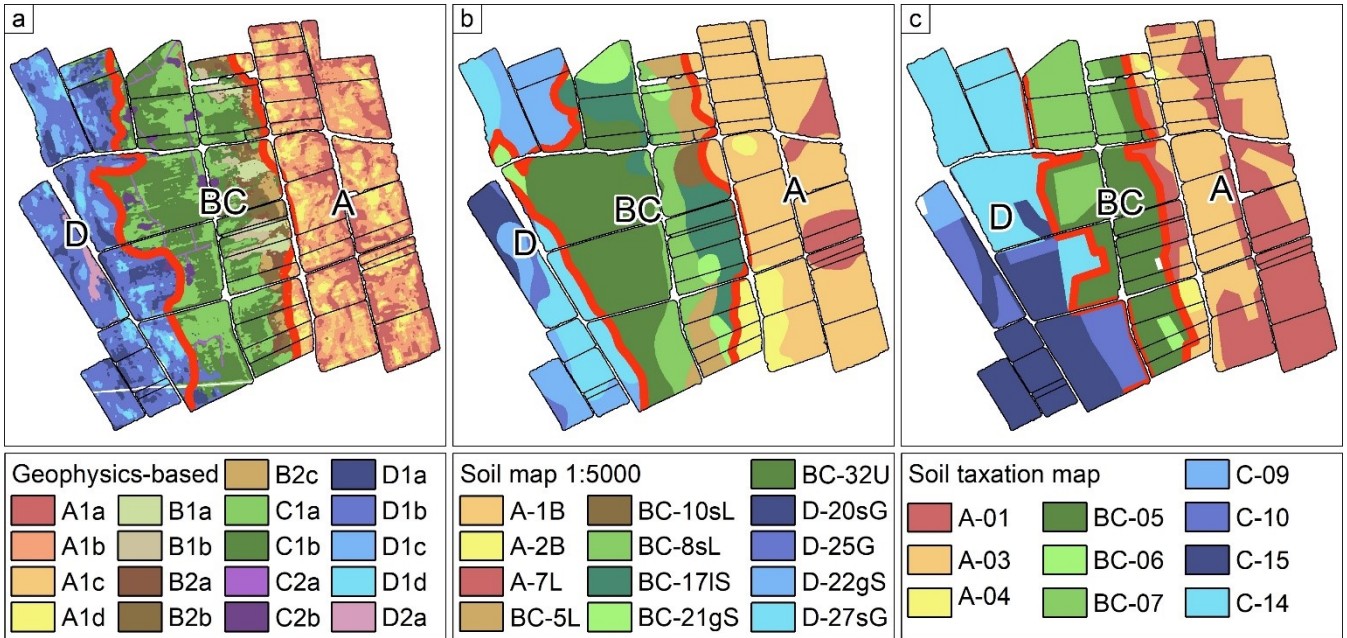

**Figure 3: Spatial distribution of soil units and of sub-areas A, BC, and D in a) the geophysics-based soil map, b) the 1:5000 soil map, and c) the soil taxation map**

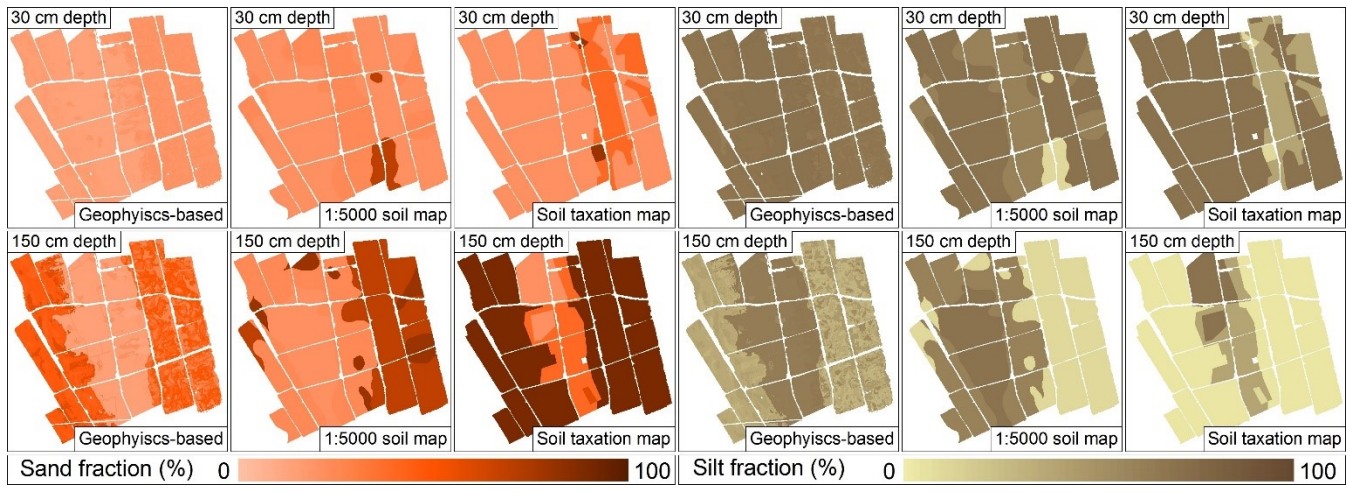


**Figure 4: Percentages of sand (orange color scale) and silt (brown color scale) at 0.30 m and at 1.50 m in the three soil maps**





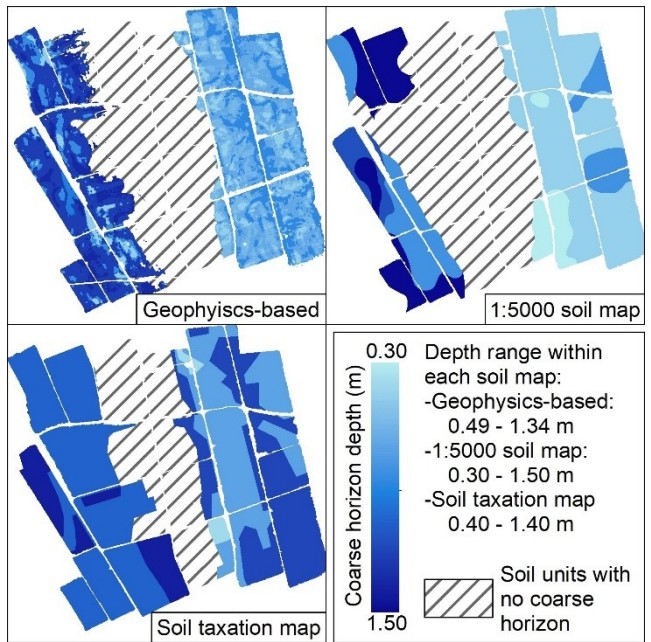

**Figure 5: Depth of the coarse horizon 2C in the three soil maps**

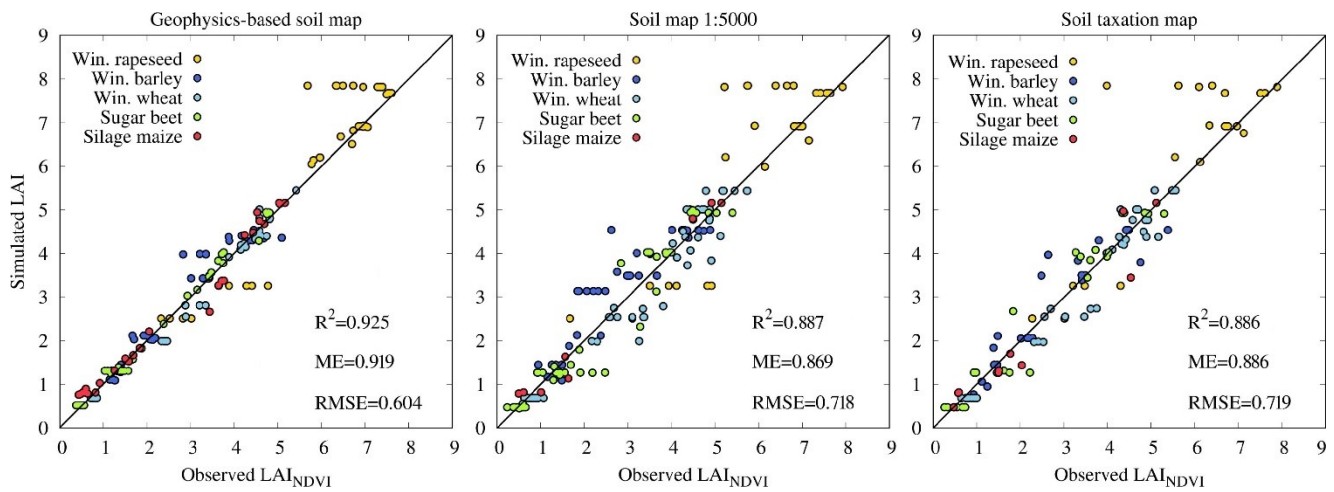


**Figure 6: Simulated LAI and mean LAI$_{NDVI}$ for each soil-crop combination for simulations based on a) the geophysics-based soil map, b) the 1:5000 soil map, and c) the soil taxation map for all six available RapidEye images.**

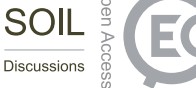

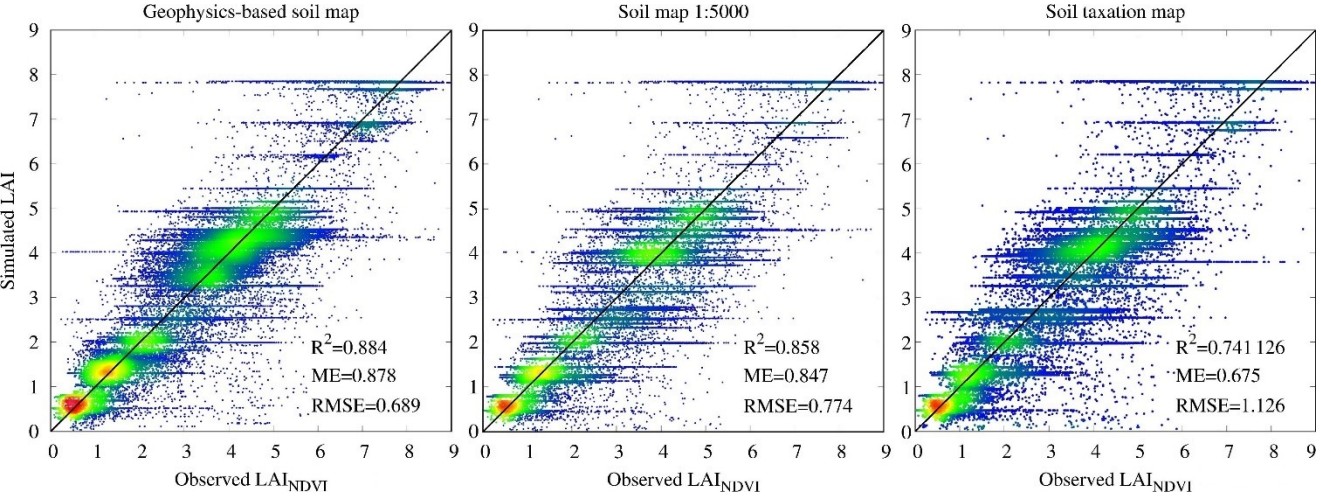

**Figure 7: Pixel-by-pixel comparison between simulated LAI and observed LAI$_{NDVI}$ for simulations based on a) the geophysics-based soil map, b) the 1:5000 soil map, and c) the soil taxation map. The color indicates the density of events, with red representing highest density and blue lowest density.**

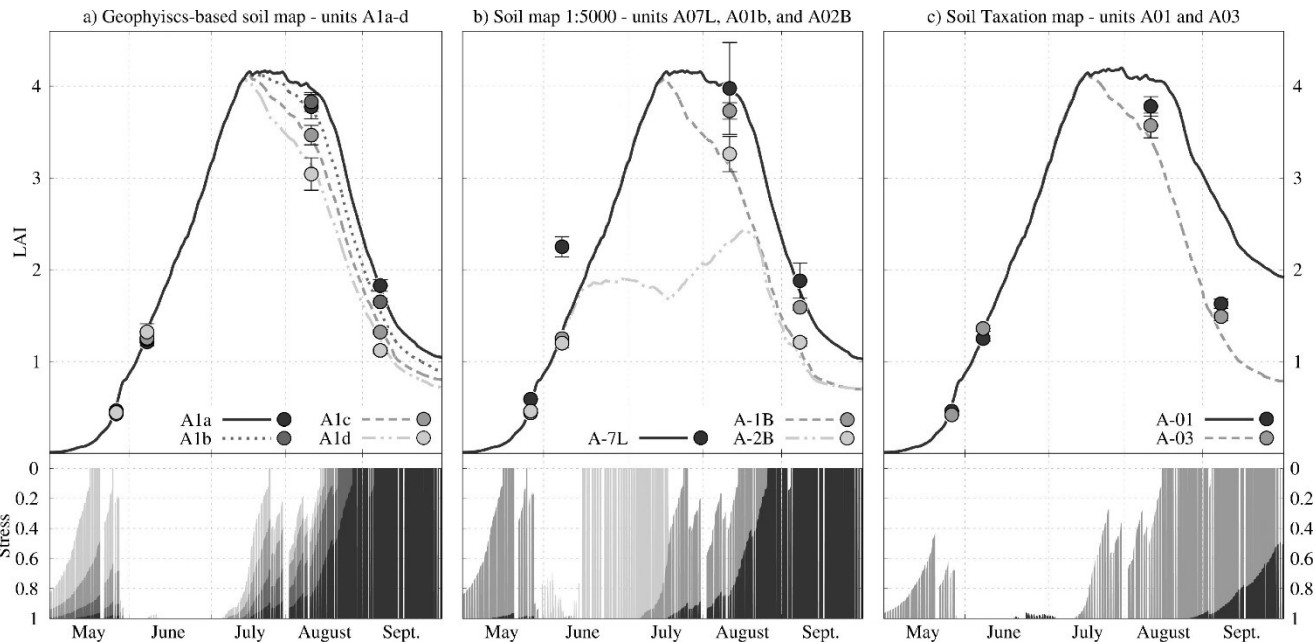

**Figure 8: Observed LAI$_{NDVI}$ (dots) of sugar beet in sub-area A compared to the LAI (lines) as well as corresponding stress occurrence simulated using input from a) the geophysics-based soil map, b) the 1:5000 soil map, and c) the soil taxation map**



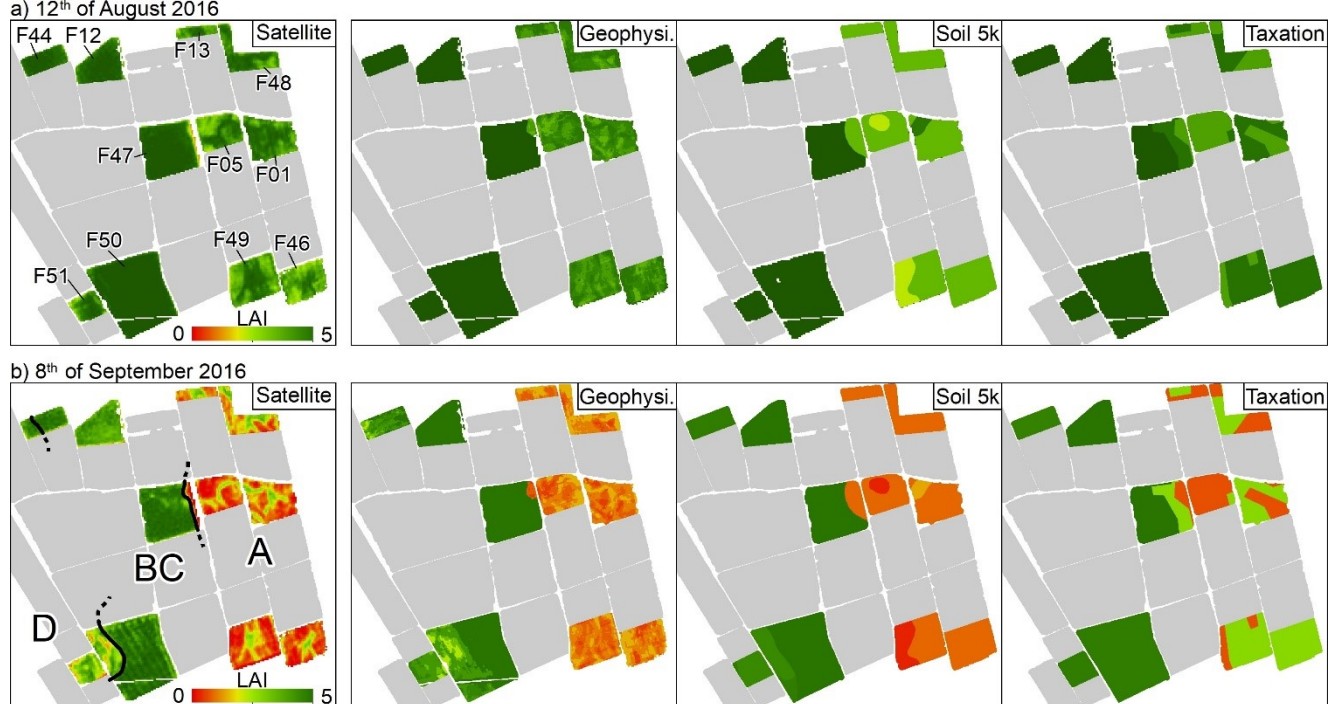

**Figure 9: Observed LAI$_{NDVI}$ and simulated LAI obtained using input from the geophysics-based soil map (Geophysi.), the 1:5000 soil map (Soil 5k), and the soil taxation map (Taxation); a) shows the codes of the investigated fields and the comparison at the 12$^{th}$ August, b) shows the geometry of sub-areas A, BC, and D and the comparison at the 8$^{th}$ September 2016**




**List of tables**

Table 1 Range of emergence and harvest dates and total area for the five crops cultivated in the study area in 2016.

| Plant | Emergence | Harvest | Area (ha) |
|---|---|---|---|
| Silage maize | 02.05.2016 | 14.11.2016 | 3.5 |
| Sugar beet | 1. to 10.05.2016 | 20.10.2016 | 26.5 |
| Winter rapeseed | 1. to 10.11.2015 | 20.07.2016 | 9.2 |
| Winter barley | 1. to 10.12.2015 | 25.07.2016 | 18.2 |
| Winter wheat | 15.11.2015 | 29.07.2016 | 26.2 |

Table 2 Unified codes of the soil units for the three soil maps.

| Soil map | Sub-area A | Sub-area BC | Sub-area D |
|---|---|---|---|
| Geophysics-based soil map | A1a, A1b, A1c, A1d | B1a, B1b, B2a, B2b, B2c, C1a, C1b, C2a, C2b | D1a, D1b, D1c, D1d, D2a |
| 1:5000 Soil map | A-1B, A-2B, A-7L | BC-5L, BC-8sL, BC-10sL, BC-17lS, BC-21gS, BC-32U | D-20sG, D- 22gS, D-25G, D-27sG |
| Soil taxation map | A-01, A-02, A-03 | BC-05, BC-06, BC-07 | D-09, D-10, D-14, D-15 |

Table 3 percentage (%) of sand, silt, and clay of the centroid of relevant soil textural classes within the USDA soil texture triangle.

| Soil class | Clay (%) | Silt (%) | Sand (%) |
|---|---|---|---|
| Loam | 20 | 40 | 40 |
| Loamy sand | 10 | 10 | 80 |
| Loamy silt | 15 | 60 | 25 |
| Sand | 5 | 5 | 90 |
| Sandy clay loam | 30 | 10 | 60 |
| Sandy loam | 20 | 10 | 70 |
| Silt | 5 | 90 | 5 |
| Silty loam | 8 | 70 | 22 |






Table 4 R², ME, and RMSE of the pixel-by-pixel comparison of the three soil maps performed between simulated LAI and observed LAI_NDVI of each soil-crop unit. The highest R² and ME and the lowest RMSE at each date are marked in bold.

| | Geophysics-based map | | | 1:5000 Soil map | | | Soil taxation map | | |
|---|---|---|---|---|---|---|---|---|---|
| Date | R² | ME | RMSE | R² | ME | RMSE | R² | ME | RMSE |
| 14th Mar | **0.84** | **0.76** | **0.62** | 0.83 | 0.75 | 0.63 | 0.72 | 0.62 | 0.79 |
| 20th Apr | 0.72 | **0.56** | **1.07** | **0.72** | 0.55 | 1.09 | 0.45 | -0.29 | 1.84 |
| 28th May | **0.93** | **0.93** | **0.64** | 0.92 | 0.92 | 0.67 | 0.81 | 0.36 | 1.14 |
| 09th Jun | 0.89 | 0.89 | **0.64** | **0.90** | **0.90** | 0.69 | 0.84 | 0.80 | 0.86 |
| 12th Aug | **0.47** | **0.47** | **0.64** | 0.39 | -0.02 | 0.89 | 0.38 | 0.37 | 0.70 |
| 08th Sep | **0.78** | **0.77** | **0.56** | 0.65 | 0.56 | 0.78 | 0.50 | 0.44 | 0.87 |

Table 5 RMSE between simulated LAI and LAI_NDVI in fields with sugar beet. The best result at each date is marked in bold.

| | 12th August | | | 8th September | | |
|---|---|---|---|---|---|---|
| Field | Geophysics-based soil map | 1:5000 Soil map | Tax. map | Geophysics-based soil map | 1:500 Soil map | Tax. map |
| F01 | **0.56** | 0.91 | 0.57 | **0.45** | 0.53 | 0.88 |
| F05 | **0.58** | 0.91 | 0.67 | **0.40** | 0.47 | 0.53 |
| F13 | **0.52** | 0.73 | 0.57 | **0.56** | 0.73 | 0.73 |
| F46 | 0.62 | **0.52** | 0.92 | **0.42** | 0.44 | 1.27 |
| F48 | **0.52** | 0.98 | 0.57 | **0.62** | 0.78 | 0.90 |
| F49 | **0.58** | 1.01 | 0.61 | **0.44** | 0.47 | 1.10 |
| F12 | **0.63** | 0.66 | 0.66 | **0.72** | 0.73 | 0.73 |
| F44 | **0.57** | 0.62 | 0.58 | 0.51 | 0.45 | **0.43** |
| F47 | 0.68 | 0.63 | **0.56** | **0.68** | 0.79 | 0.77 |
| F50 | 0.69 | **0.65** | 0.66 | **0.51** | 0.76 | 0.79 |
| F51 | **1.21** | 1.24 | 1.22 | **0.95** | 1.19 | 1.32 |