# Peer review of "Added value of geophysics-based soil mapping in agro-ecosystem simulations"

_SOIL, 2020_

## Referee Comment (RC1) · Anonymous Referee #1 · 26 Jan 2021

General comments This paper investigates the added value of geophysics-based soil mapping in agro-ecosystem simulations. This is an interesting work that provides insights on how geophysics-based soil mapping can help improve the outputs of agro-ecosystems simulations. It is well written and the results are of good quality. However, I have some comments below that I think would help increase the quality of the paper and the interest of potential readers of the SOIL journal.

Specific comments First, the authors evaluated the added value of the geophysics-based soil mapping based on crop leaf area index (LAI) simulations and generated interesting results. However, I wonder why in this paper the authors did not simulate

the crop yield as an additional parameter. I think that it would be of great interest to the reader to assess the added value of the geophysics-based soil mapping on the simulation of the yield of the studied crops using the agro-ecosystem model AgroC. Results with crop yields would help determine e.g. if the added-value of geophysics-based soil mapping could be economically interesting to potential farmers. Sugar beet would be a good case study for yield simulations with the AgroC model as it represents 31.7% of the study area.

Second, the "Conclusion" section is just a summary (repetition) of the key results and provides no perspectives for future research. I am quite convinced that readers of the SOIL journal would be interested to know what would be the next step(s).

Technical corrections L12: "km2" instead of "km2"

L14: "R2" instead of "R2"

L19: ". . .; Stafford"

L19: ". . .; Sylvester-Bradley"

L21: ". . .; Sylvester-Bradley"

L21: ". . .; Chartzoulakis"

L26: ". . .; Nussbaum"

L26: ". . .; Pätzold et al., 2008"

etc. Please check the above typos through the whole text.

L65-66: Could the authors name those few published studies that linked geophysics-based soil maps and crop growth model?

L74, L81, L465: "1 x 1 km" or "1 km x 1 km"?

L81: Could the authors provide the geographical coordinates of the study area?

L97-98 and similar: top soil or topsoil?

L121: What is the scale of the national soil and yield potential map (NRW, 1960)?

L160-162: Does the description of the four sub-areas (A, B, C and D) refer to a previous work? If so, please provide the references.

L163: I would suggest: "...as a single sub-area BC in this study (Fig. 3)."

L186: Is $\alpha$ the inverse of the air entry pressure or a parameter related to the inverse of the air entry pressure?

L190: The Q term has already been described in L179-180.

L198: "...in dependence of pressure head" or "...dependent of the pressure head"?

L200: The "=" sign next to ÏŢ(h) is missing in Equation 7. Moreover, Equation 7 needs to be much more clearly written to avoid confusion.

L208: "...using pedotransfer functions (PTFs)" instead of "...using a pedotransfer function (PTF)"

L234-235: "the PTFs of..." instead of "the pedotransfer function of..."

L238: "where" instead of "were"

L242: "...is of primary importance" or "...are of primary importance" (if you refer to both the coarse sand and the gravel horizon 2C)

L255-256: Why starting from 1st of January 2015 instead of 1st of July 2015 for the spin-up simulation?

L267: "leaves" or "leafs"?

L279: The authors should mention what do "Obsi,", "Simi" and "n" refer to (see Equation 10).

L300: "This is a consequence of" instead of "This is as a consequence of"

L310: "values" instead of "vales"

L318: I would suggest: "...topsoil (at 0.30 m depth) in the three..."

L324: I would suggest: (results not shown)

L328, Figure 5: I cannot tell the difference between "soil layer thickness" and "depth" in Figure 5.

L330-331, Figure 5: The figures of the depth range in the text and in Figure 5 are a bit different: 0.47-1.34 m (in the text) vs 0.49-1.34 m (in Figure 5) for the geophysics-based soil map; 0.30- 1.40 m (in the text) vs 0.40-1.40 m (in Figure 5) for the soil taxation map.

L332-334: I am missing something here. It seems to me that the depths (range) of the coarse horizon in the 1:5000 regional soil map (i.e. 0.30-1.50 m in Figure 5) were closer to that of the soil taxation map (i.e. 0.40-1.40 m in Figure 5) compared to the geophysics-based soil map. Please could you confirm?

L354: For sake of uniformity, I would suggest: "mean LAINDVI" instead of "average LAINDVI"

L368: I would suggest: "great water stress" instead of "strong water stress"

L374: I would suggest: "...was the greatest" instead of "...was strongest"

L382: "0.58 m in A1c" instead of "0.58 in A1c"

L385: "matched well the observations" or "well matched the observations"?

L393: I would suggest: "a greater water stress" instead of "a stronger water stress

L396-397: It seems to me that the error bars in Figure 8c rather show low variability within each unit compared to the error bars in Figure 8b. Please could you confirm?

L407: "LAINDVI reached"

L420: "results" instead of "result"

L425: "based on the RMSE" instead of "based on to the RMSE"

L425: "consists" instead of "consist"

L427: I would suggest "for both dates" instead of "for both days"

L459: "this reduction in performance was caused..." instead of "this reduction was caused..."

L464: "Conclusion" instead of "Conclusions"

L689: 1:5000 instead of "1.5000"

L695, Figure 4: I would suggest: "...at 0.30 m depth and at 1.50 m depth in the..." instead of "...at 0.30 m and at 1.50 m in the..."

L700, Figure 6: I would suggest "Mean observed LAINDVI" instead of "Observed LAINDVI" as the X-axis title.

L704, Figure 7: "R2= 0.741" (i.e. 3 digits) instead of "R2= 0.741126" (i.e. 6 digits)

L705-706, Figure 7: "...representing the highest density and blue the lowest density."

L709, Figure 8: I would suggest: "the simulated LAI (lines)" instead of "the LAI (lines)"

L710, Figure 8: d, e, f are not mentioned in Figure 8 and are not referred in the caption

L713-714: "on 12th August 2016" instead of "at the 12th August"

L714, Figure 9: "on 8th September 2016" instead of "at the 8th September 2016"

L730, Table 5: "1:5000 Soil map" instead of "1:500 map"

L731, Table 5: I would suggest "12th August 2016" and "8th September 2016"

---

## Short Comment (SC1) · 1 Feb 2021

This paper "Added value of geophysics-based soil mapping in agro-ecosystem simulations" shows interstitium improvements in combining geophysically based soil mapping with agro-ecosystems modelling. It is well written and suits with the scope of the SOIL Journal. We agree with the previous comments of the first reviewer and only a have one more specific comment:

L60: refers to the work "Determining the within-field yield variability. . ." using EMI based soil maps. Actually, the mobile measurement system in this work uses electrical resistivity mapping/tomography (ERT), which is a specific type of measurement using direct

current measurements rather than magnetic induced ones. We suggest to rephrase the sentence as "Some case studies successfully used EMI and ERT-based soil maps. . .". You may also leave out both terms 'EMI' and 'ERT' and merely refer to "geophysically based soil maps".

---

## Referee Comment (RC2) · Jacopo Boaga (Referee) · 3 Feb 2021

The paper regards the comparison of geophysics-based and classical soil mapping as input for agronomical simulations. I agree with the previous revisions of the colleagues. Paper is well written and results are supported by the data presented. Topic is of interest for Soil and I think the paper is acceptable for publication after minor corrections.

In particular authors should underlined that nowadays inversion of Eca data are common, since a number of proficient codes are available. From the paper reading it seems only nominal exploration depth of EMI is used, limiting the precision of the subsoil characterisation. This is just one way to use EMI survey, but not the unique one. Inversion

routine suitable for the equipment used for the study is easily accessible (e.g. EmgapY by McLachlan et al., just to name one). This option should be at least suggested in the text, as the resolution limits of the technique and the problems that can occur with calibration/drift (e.g. Von Hebel et al 2019, Mester et al. 2014, Boaga 2017, Tan et al. 2018).

Ln54-56 This sentence about management is not clear, please re-phrase

Fig.2a Please provide a color scale and improve caption (also in other figures). A reader should have a clear comprehension of a figure just from the caption.

Ln310 'vales' instead of 'values'?

FIg. 7 insert a color scale for the density

Ln 355 please introduce RapidEye for non specialist

Fig.8 improve caption

Ln 395-400 Specify if you are using the nominal exploration depth of EMI

Thanks for the very interesting contribution it was a pleasure to read
* * *

---

## Author Comment (AC1) · 3 Feb 2021

Dear Eric Bönecke,

Thank you very much for your comment and for the appreciation of our work. Indeed you raise a good point. We will improve the manuscript to correct this inaccuracy and reference your work in the way it deserves.

The new version of this paragraph will read: "Some case studies where geophysics-based soil information were successfully combined with crop growth models are available (Boenecke et al., 2018; Wong and Asseng, 2006)."

This might change slightly depending on further reviewers requests, but the scientific meaning of the sentence and the way your work is referenced will not be changed.

Thanks again for taking the time to read our work.

Best regards, Cosimo Brogi (on behalf of all co-authors)
* * *

---

## Author Comment (AC2) · 7 Feb 2021

We thank the anonymous referee for taking the time to evaluate our manuscript. The provided review is nice and thorough and I can assure you that all comments will be carefully addressed.

Once the open discussion of the manuscript will come to an end, we will provide a new edited version of the manuscript and a response letter where comments and edits from both referees as well as other eventual sources will be addressed and discussed in detail.

[Figure]

A couple of aspects from this review can be already mentioned here in this short and preliminary reply:

First, we will try to include data on crop productivity for sugar beet. Unfortunately, we have very limited actual yield data from the given year in the study area. Thus, only one small field cropped with sugar beet has the necessary data. Furthermore, it is true that the yield simulated with the three maps can be compared. However, to provide farmers with a meaningful economical evaluation it would be necessary to investigate in detail further aspects such as irrigation and fertilization (preferably across multiple seasons and with different climatic variables). Thus, we would like to stress that this would go well beyond the scope of this study. Another aspect that should be taken into account is that LAI, for which we have a nice dataset, is more and more frequently used as a proxy for biomass in crop modelling (as the two are generally closely linked). For these reasons, we think that including the proposed information on yield will be an interesting addition, but we are aware that we do not have the data to expand the topic to the broad boundaries suggested by the reviewer. We will then carefully introduce the topic and discuss the limitations of our findings as well as the aspects mentioned above and the potential for future research.

Second, editing parts of the conclusions is a very good suggestion that we will be happy to implement. This could not only discuss future research but also aspects such as the one proposed by the referee in his previous comment in addition to those brought up by the second referee.

Thank you again for your time, Cosimo Brogi, on behalf of all co-authors

———————————————————

---

## Author Comment (AC3) · 7 Feb 2021

Dear Jacopo Boaga,

Thank you for the time you invested to evaluate our manuscript and for the comments that will be carefully addressed.

Once the open discussion of the manuscript will come to an end, we will provide a new edited version of the manuscript and a response letter where comments and edits from both referees as well as other eventual sources will be addressed and discussed in detail.

[Figure]

We will also work on expanding the descriptions and details of the methodology that was used to produce the EMI-based soil map. This is described in detail in a previous publication to which we often refer to in the manuscript (Brogi et al. 2018 Geoderma), but we will carefully go through the manuscript to verify that all the necessary information are given to the reader. We will also include the possibility that future studies could make use of inverted EC maps when rewriting parts of the conclusions as suggested by the first referee. However, we hope you will understand if these parts will not be long-winded as the geophysics-based product that we use was produced in a previous separated study and we do not apply meaningful changes to such geophysical approach.

Thank you again for your time, Cosimo Brogi, on behalf of all co-authors

---

## Author Comment (AC4) · 17 Feb 2021

*Dear Editors and Referees,*

*We appreciate the time and effort that you have invested to evaluate our manuscript.*

*We have carefully read, addressed, and provided replies (in cursive) to all the comments from the interactive discussion and we think that this has improved the manuscript considerably.*

*In this new version of the manuscript, among other improvements, we added an analysis of yield simulations for one selected field (as this was the only one with the neces-*

*sary data) and we introduced additional outlook aspects in the conclusion section, one being the use of inverted EMI datasets.*

*We hope that the manuscript can now be accepted for publication in SOIL.*

*Best Regards,*

*Cosimo Brogi (on behalf of all co-authors)*

**Referee comment 1 (Anonymous)**

General comments:

This paper investigates the added value of geophysics-based soil mapping in agro-ecosystem simulations. This is an interesting work that provides insights on how geophysics-based soil mapping can help improve the outputs of agroecosystems simulations. It is well written and the results are of good quality. However, I have some comments below that I think would help increase the quality of the paper and the interest of potential readers of the SOIL journal.

*REPLY: We thank the referee for the positive and constructive feedback for our manuscript.*

Specific comments: First, the authors evaluated the added value of the geophysics-based soil mapping based on crop leaf area index (LAI) simulations and generated interesting results. However, I wonder why in this paper the authors did not simulate the crop yield as an additional parameter. I think that it would be of great interest to the reader to assess the added value of the geophysics-based soil mapping on the simulation of the yield of the studied crops using the agro-ecosystem model AgroC. Results with crop yields would help determine e.g. if the added-value of geophysics-based soil mapping could be economically interesting to potential farmers. Sugar beet would be a good case study for yield simulations with the AgroC model as it represents 31.7% of the study area.

*REPLY: We thank the referee for the appreciation of our work and understand the point raised here. We agree that a comprehensive analysis of the yield simulation alongside LAI simulation would be very interesting and useful. However, the data on actual yield recorded by the farmers for the study area is rather limited. In fact, we have available yield data only for one field (F01) cropped with sugar beet. Nonetheless, encouraged by the referee comment, we now included an analysis of simulated yield that can be found towards the end of section 4.3. The revised text now reads:*

*"In field F01 of sub-area A, the reduction of the RMSE between LAI simulations and observations provided by the use of the geophysics-based soil map varied between 15% and 49% at the 8th of September. In this field, the yield recorded by the farmer was 14.2 t ha⁻¹ of dry beet biomass. The use of inputs from the geophysics-based soil map resulted in a simulated yield of 14.3 t ha⁻¹, which was only 1% higher than actual yield. The yield simulated with inputs from the commonly available map over-estimated yield with 15.4 t ha⁻¹ for the 1:5000 soil map (+8%) and 17.3 t ha⁻¹ for the soil taxation map (+22%). Similar as in the case of LAI, the geophysics-based soil map outperformed the other two maps. However, these results should be interpreted carefully as the required information was available only for a rather small field within the study area. Nonetheless, it is anticipated that the geophysics-based soil map provides similar improvements in yield estimates as for the LAI simulations for other fields. This expectation is based on the tight relationship between LAI and other aspects of crop development, such as biomass production and ultimately yield. For this same reason, the assimilation of LAI data into crop growth model ensembles have been shown to improve yield estimates (Jonckheere et al., 2004; Tewes et al., 2020; Wilhelm et al., 2000)".*

*In this new section, we tried to make the best out of the small amount of available yield data. We also considered the possibility to extend the yield analysis to all fields cropped with sugar beet or to the entire study area. However, we concluded that such an approach is not recommended as the different distribution of soil units with different*

*soil characteristics within the various fields makes it impossible to precisely estimate the actual yield starting from the available dataset. Therefore, we think that the new paragraph and analysis that we provided make good use of the data that we have available without jumping to conclusions that are not supported by data*

Second, the "Conclusion" section is just a summary (repetition) of the key results and provides no perspectives for future research. I am quite convinced that readers of the SOIL journal would be interested to know what would be the next step(s).

*REPLY: We thank the reviewer for this suggestion and we agree that the conclusion section can be improved. Following this suggestion, we shortened some parts of the conclusions and included a final paragraph on the possible outlook of this study. In doing this, we also incorporated the comments of the second referee on certain aspects of geophysical data analysis. The last paragraph of the conclusion now reads:*

*"Nevertheless, the strong dependence of the added value of the geophysics-based soil map on the crop type, soil characteristics, and precipitations calls for further research on such topics. For example, it would be important to estimate the added value of a geophysics-based soil map before this soil map is realized, so that farmers could make an informed decision on the profitability of such a mapping product. This could be achieved by investigating the specific pedoclimatic conditions of a given farm or by mapping small portions of the target area to provide an estimate of the costs and benefits of the final soil map. Similarly, by performing long-term simulations, farmers could be informed on the time span required to achieve returns on their investment. Such simulations should include agricultural practices that are key to certain areas such as irrigation, crop rotation, and different fertilization practices. In parallel, the use of different geophysical techniques (taken singularly or in combination) and their added value should be further investigated. In the case of electromagnetic induction (EMI), focus should be put on the different added value for agro-ecosystem models originating from rather simple apparent electrical conductivity (ECa) maps or from more refined products based on the calibration and inversion of ECa measurements".*

Technical corrections:

L12: "km$^2$" instead of "km2"

*REPLY: Following the reviewer suggestion, the proposed comment was implemented in this new version of the manuscript.*

L14: "R$^2$" instead of "R2"

*REPLY: The proposed comment was implemented in this new version of the manuscript.*

L19: "...; Stafford", L19: "...; Sylvester-Bradley", L21: "...; Sylvester-Bradley", L21: "...; Chartzoulakis", L26: "...; Nussbaum", L26: "...; Pätzold et al., 2008" etc. Please check the above typos through the whole text.

*REPLY: The reference style was improved in this new version of the manuscript.*

L65-66: Could the authors name those few published studies that linked geophysics-based soil maps and crop growth model?

*REPLY: Following the comment of the reviewer, we realized that the phrasing of this section was rather confusing and too long. Thus, the final part of the paragraph was rephrased and now reads: "Some case studies where geophysics-based soil information were successfully combined with crop growth models are available (Boenecke et al., 2018; Wong and Asseng, 2006). Recently, Brogi et al. (2020) successfully used inputs from an EMI-derived soil map to simulate soil water content dynamics and their effects on the growth of six crop types for a 90 ha study area. Furthermore, Krüger et al. (2013) showed that the consideration of soil depth derived from geophysical measurements improved the simulation of biomass production on a 4.4 ha experimental field compared to simulations based only on commonly available soil maps. Despite these promising results, there is need for further studies that link geophysics-based soil products and crop growth models. Moreover, the quantification of the added value of geophysics-based soil characterization for agro-ecosystem modelling applications has*
*not been thoroughly investigated yet (Krüger et al., 2013), especially in areas larger than the field-scale and for multiple crop and soil types".*

L74, L81, L465: "1 x 1 km" or "1 km x 1 km"?

*REPLY: Following the reviewer suggestion, the unit has been added to the first distance as well.*

L81: Could the authors provide the geographical coordinates of the study area?

*REPLY: Following the suggestion of the referee, the coordinated were added in the first sentence of the paragraph.*

L97-98 and similar: top soil or topsoil?

*REPLY: Following the suggestion of the referee, the wording was changed throughout the manuscript.*

L121: What is the scale of the national soil and yield potential map (NRW, 1960)?

*REPLY: The scale of this map can vary depending on the region. For this, we added this detail in a later paragraph where this map is described. The new sentence reads: 'The scale of this map can vary across different region in Germany and is 1:5000 for the study area.'*

L160-162: Does the description of the four sub-areas (A, B, C and D) refer to a previous work? If so, please provide the references.

*REPLY: Following this suggestion, we added a reference to the proper study describing the for sub-areas in more detail.*

L163: I would suggest: "...as a single sub-area BC in this study (Fig. 3)."

*REPLY: Following this suggestion, we added the indication of the relevant figure.*

L186: Is $\alpha$ the inverse of the air entry pressure or a parameter related to the inverse of the air entry pressure?

*REPLY: Following this comment, we now specify that "$\alpha$ is a parameter corresponding approximately to the inverse of the air entry pressure (cm-1)".*

L190: The Q term has already been described in L179-180.

*REPLY: This sentence was removed following this comment.*

L198: "...in dependence of pressure head" or "...dependent of the pressure head"?

*REPLY: This sentence was rephrase following the comment of the referee and now reads 'the water uptake is scaled by a water uptake stress factor $\varphi$ that depends on the pressure head'.*

L200: The "=" sign next to Ï ÂÿT(h) is missing in Equation 7. Moreover, Equation 7 needs to be much more clearly written to avoid confusion.

*REPLY: Details were added to the equation while prescribed thresholds were moved from the equation to text improve the readability. The new text after the equation reads: "where h0-3 (cm) are threshold pressure heads obtained from literature. These thresholds were set to h0 = 0 cm, h1 = −20 cm, h2 = −5000 cm, and h3 = −16000 cm (Vanclooster et al., 1995)".*

L208: "...using pedotransfer functions (PTFs)" instead of "...using a pedotransfer function (PTF)"

*REPLY: The proposed comment was implemented in this new version of the manuscript.*

L234-235: "the PTFs of..." instead of "the pedotransfer function of..."

*REPLY: The proposed comment was implemented in this new version of the manuscript.*

L238: "where" instead of "were"

*REPLY: The proposed comment was implemented in this new version of the*

*manuscript.*

L242: "...is of primary importance" or "...are of primary importance" (if you refer to both the coarse sand and the gravel horizon 2C)

*REPLY: Following the comment of the referee, we rephrased this sentence to avoid confusion since the coarse horizon 2C is composed of sand and gravel. The new version of the manuscript provides an improved phrasing which reads: "the coarse horizon 2C (sand and gravel) that underlies the fine aeolian sediments in sub-area A and in parts of sub-area D is of primary importance".*

L255-256: Why starting from 1st of January 2015 instead of 1st of July 2015 for the spin-up simulation?

*REPLY: We thank the reviewer for this comment. The spin up was run between the start of the year 2015 and the end of the year 2016 to complete the simulation of two years and thus two annual cycles. If the spin-up would have been started in July, the initial pressure head in the soil column would not be comparable with the final values obtained in December. Therefore, we prefer to start the spin-up in January. As model spin-up is a common practice and as the manuscript is already quite long, we did not think that it is necessary to add further details and considerations in the manuscript.*

L267: "leaves" or "leafs"?

*REPLY: The proposed comment was implemented in this new version of the manuscript.*

L279: The authors should mention what do "Obsi,", "Simi" and "n" refer to (see Equation 10).

*REPLY: Such description was implemented in this new version of the manuscript and is located immediately after Equation 10.*

L300: "This is a consequence of" instead of "This is as a consequence of"

*REPLY: The proposed comment was implemented in this new version of the manuscript.*

L310: "values" instead of "vales"

*REPLY: The proposed comment was implemented in this new version of the manuscript.*

L318: I would suggest: "...topsoil (at 0.30 m depth) in the three..."

*REPLY: The proposed comment was implemented in this new version of the manuscript.*

L324: I would suggest: (results not shown)

*REPLY: The proposed comment was implemented in this new version of the manuscript.*

L328, Figure 5: I cannot tell the difference between "soil layer thickness" and "depth" in Figure 5.

*REPLY: Here we referred to the thickness and depth of all the soil layers that are present in the profiles. Some differences are present for all layers, but the differences are much more apparent for the deeper coarse sand and gravels (when present). However, we understand that this formulation can cause confusion and that the aim of the paragraph (and of figure 5) is to tackle the depth of the coarse sand and gravel. Thus, the new version of the manuscript reads: "Differences in the depth to the coarse sand and gravel that underlies the fine sediments in sub-areas A and D were observed between the three soil maps (Fig. 5)".*

L330-331, Figure 5: The figures of the depth range in the text and in Figure 5 are a bit different: 0.47-1.34 m (in the text) vs 0.49-1.34 m (in Figure 5) for the geophysics-based soil map; 0.30-1.40 m (in the text) vs 0.40-1.40 m (in Figure 5) for the soil taxation map.

*REPLY: We thank the referee for this comment. Figure 5 was edited to include the right*

*depth range (0.47 – 1.34 m for the geophysics-based soil map) and the typo in text in the range of the soil taxation map (now 0.40 and 1.40 m) has been corrected.*

L332-334: I am missing something here. It seems to me that the depths (range) of the coarse horizon in the 1:5000 regional soil map (i.e. 0.30-1.50 m in Figure 5) were closer to that of the soil taxation map (i.e. 0.40-1.40 m in Figure 5) compared to the geophysics-based soil map. Please could you confirm?

*REPLY: We again thank the referee for this comment. After taking into account the corrections applied after the previous comment (L330-331) , the text has been edited and now reads: 'Both the two commonly available soil maps showed a larger depth range that that of the geophysics-based soil map with the 1:5000 soil map showing the largest depth range".*

L354: For sake of uniformity, I would suggest: "mean LAINDVI" instead of "average LAINDVI"

*REPLY: The proposed comment was implemented in this new version of the manuscript.*

L368: I would suggest: "great water stress" instead of "strong water stress"

*REPLY: Following the comment of the referee we decided to use "high" and "low" water stress throughout the manuscript.*

L374: I would suggest: "...was the greatest" instead of "...was strongest"

*REPLY: The proposed comment was implemented in this new version of the manuscript.*

L382: "0.58 m in A1c" instead of "0.58 in A1c"

*REPLY: The proposed comment was implemented in this new version of the manuscript.*

L385: "matched well the observations" or "well matched the observations"?

*REPLY: The proposed comment was implemented in this new version of the manuscript.*

L393: I would suggest: "a greater water stress" instead of "a stronger water stress

*REPLY: The proposed comment was implemented in this new version of the manuscript.*

L396-397: It seems to me that the error bars in Figure 8c rather show low variability within each unit compared to the error bars in Figure 8b. Please could you confirm?

*REPLY: We thank the reviewer for this comment. In fact, figure 8 did not carry the exact data for plot c and this has been now corrected. Now, the error bars are the actual ones and the variability described in the text is apparent in the figure as well. Thus, the main text was not changed after correcting figure 8.*

L407: "LAINDVI reached"

*REPLY: The proposed comment was implemented in this new version of the manuscript.*

L420: "results" instead of "result"

*REPLY: The proposed comment was implemented in this new version of the manuscript.*

L425: "based on the RMSE" instead of "based on to the RMSE"

*REPLY: The proposed comment was implemented in this new version of the manuscript.*

L425: "consists" instead of "consist"

*REPLY: The proposed comment was implemented in this new version of the manuscript.*

L427: I would suggest "for both dates" instead of "for both days"

*REPLY: The proposed comment was implemented in this new version of the manuscript.*

L459: "this reduction in performance was caused..." instead of "this reduction was caused..."

*REPLY: The proposed comment was implemented in this new version of the manuscript.*

L464: "Conclusion" instead of "Conclusions"

*REPLY: According to the 'manuscript composition' section of the instruction for submission in SOIL, this section should be called 'conclusions'. Thus, the text was not edited here.*

L689: 1:5000 instead of "1.5000"

*REPLY: The proposed comment was implemented in this new version of the manuscript.*

L695, Figure 4: I would suggest: "...at 0.30 m depth and at 1.50 m depth in the..." instead of "...at 0.30 m and at 1.50 m in the..."

*REPLY: The proposed comment was implemented in this new version of the manuscript.*

L700, Figure 6: I would suggest "Mean observed LAINDVI" instead of "Observed LAINDVI" as the X-axis title.

*REPLY: The figure was edited to follow the suggestion of the reviewer.*

L704, Figure 7: "R2= 0.741" (i.e. 3 digits) instead of "R2= 0.741126" (i.e. 6 digits)

*REPLY: The figure was corrected to show consistent use of digits.*

L705-706, Figure 7: "...representing the highest density and blue the lowest density."

*REPLY: The proposed comment was implemented in this new version of the manuscript.*

L709, Figure 8: I would suggest: "the simulated LAI (lines)" instead of "the LAI (lines)"

*REPLY: The proposed comment was implemented in this new version of the manuscript.*

L710, Figure 8: d, e, f are not mentioned in Figure 8 and are not referred in the caption

*REPLY: Based on the comment of the reviewer, we understood that the text is not fully clear at this point. The lower plots with water stress are part of plots a, b, and c. To better describe this in the caption, we now included notations for (upper plots) and (lower plots) in the text so that the readability is improved.*

L713-714: "on 12th August 2016" instead of "at the 12th August"

*REPLY: The proposed comment was implemented in this new version of the manuscript.*

L714, Figure 9: "on 8th September 2016" instead of "at the 8th September 2016"

*REPLY: The proposed comment was implemented in this new version of the manuscript.*

L730, Table 5: "1:5000 Soil map" instead of "1:500 map"

*REPLY: The proposed comment was implemented in this new version of the manuscript.*

L731, Table 5: I would suggest "12th August 2016" and "8th September 2016"

*REPLY: The proposed comment was implemented in this new version of the manuscript.*

**Referee comment 2 (Jacopo Boaga)**

The paper regards the comparison of geophysics-based and classical soil mapping as input for agronomical simulations. I agree with the previous revisions of the colleagues. Paper is well written and results are supported by the data presented. Topic is of interest for Soil and I think the paper is acceptable for publication after minor corrections.

In particular authors should underlined that nowadays inversion of ECa data are common, since a number of proficient codes are available. From the paper reading it seems only nominal exploration depth of EMI is used, limiting the precision of the subsoil characterization. This is just one way to use EMI survey, but not the unique one. Inversion routine suitable for the equipment used for the study is easily accessible (e.g. EmgapY by McLachlan et al., just to name one). This option should be at least suggested in the text, as the resolution limits of the technique and the problems that can occur with calibration/drift (e.g. Von Hebel et al 2019, Mester et al. 2014, Boaga 2017, Tan et al. 2018).

*REPLY: We thank the referee for the comment and we carefully went through our manuscript to identify how to address it. The data analysis behind the creation of the geophysics-based soil map is discussed in detail in Brogi et al. (2018), which also discusses the limitations of the dataset and of the methods. The geophysics-based soil map used here was identical to the published map in Brogi et al. (2018). Thus, this map is only shortly described in our manuscript. Nonetheless, in this new version of the manuscript, we clarify that the map is produced from apparent electrical conductivity (ECa) maps and we now clearly point the reader to the above mentioned publication for information on some aspects regarding data analysis (see paragraph "2.2 available soil maps"). In Brogi et al. (2018), the possibility to use inverted data as well as calibration issues are described in detail. We do not think that it is necessary to introduce these topics at this point of the manuscript as a comparison between inverted and non-inverted (or calibrated and non-calibrated) EMI mapping goes beyond the scope of the study. Furthermore, the geophysics-based soil map that was used in this study uses*

*a classification technique and a combination with direct soil sampling, and thus goes beyond the use of the depth of investigation only. In fact, the depths of the soil layers discussed throughout the manuscript are a result of the soil sampling which is then distributed in space with ECa maps. We think that this is sufficiently described in our manuscript and again, we do not think that it is necessary to add an additional level of detail as all these considerations are already explored in detail in Brogi et al (2018).*

*We considered the possibility to extensively introduce inversion methods in the introduction of this manuscript. However, this would result in a much longer introduction and in a break of the text flow. Finally, we edited the text where the novel multi-coil instruments are mentioned and shortly introduced the inversion methods. The new text reads: "However, there is renewed interest in this technique because of the development of multi-coil instruments that allow to investigate multiple depths simultaneously (Monteiro Santos et al., 2010; Saey et al., 2012; von Hebel et al., 2014) and thus enables the inversion of EMI data by combining such depths (Boaga, 2017; Mester et al., 2014; Tan et al., 2019; Von Hebel et al., 2019)". We believe that this addition provides sufficient benefit to the reader as the geophysics-based soil map that we used is not based on inversion strategies.*

*Furthermore, we agree that this topic would be a very nice addition to the conclusion section, which has been modified accordingly. In order to address this comment, the text now reads: "In parallel, the use of different geophysical techniques (taken singularly or in combination) and their added value should be further investigated. In the case of electromagnetic induction (EMI), focus should be put on the different added value for agro-ecosystem models originating from rather simple apparent electrical conductivity (ECa) maps or from more refined products based on the calibration and inversion of ECa measurements".*

Ln54-56 This sentence about management is not clear, please re-phrase

*REPLY: Following the comment of the reviewer the sentence has been improved to*

clarify the message. The new text now reads: "One drawback of EMI-derived soil maps is that they can only determine the geometry of potential management zones without directly providing information on the appropriate management of such areas (King et al., 2005)".

Fig.2a - Please provide a color scale and improve caption (also in other figures). A reader should have a clear comprehension of a figure just from the caption.

*REPLY: The color scale of this particular figure was kept equal to the original products. The scope was to show the soil maps in the same form in which they are available to the public. Colors and codes for each single soil units are provided in the following Figure 3. The addition of such coloring and legends to Figure 2 would result in a too complex image that would not be clear to the reader. Thus the use of two separate images. At the same time, the addition of the descriptions of each soil unit in figure 3 would overwhelm the reader with information that is not required to understand this study. Nonetheless, to improve the caption and the readability and following the comment of the referee, we now point the reader to Figure 3 at the end of the caption of Figure 2.*

Ln310 'vales' instead of 'values'?

*REPLY: The proposed comment was implemented in this new version of the manuscript.*

FIg. 7 insert a color scale for the density

*REPLY: We tried to introduce a color bar on the side of the figure. However, we noticed that this reduced the size of the three plots. As plots of Figure 7 are of the same dimension and scale as the plots of Figure 6 (for the purpose of comparing them), we are of the idea that the comparable size between the two figures is more important compared to having a color bar for density in this specific case. Furthermore, the density is described in the caption and uses colors that are commonly used in scientific publications. Thus, following the tests images we produced, we are of the idea that the*

*original figure is sufficiently intuitive and detailed in combination with the caption.*

Ln 355 please introduce RapidEye for non-specialist

*REPLY: we thoroughly checked our manuscript following the comment of the referee. We now added further details in section "2.1 Study area" to better inform the reader and we now refer to "RapidEye satellite image(ry)" when this topic is picked up later in the text.*

Fig.8 - Improve caption

*REPLY: The proposed comment was implemented in this new version of the manuscript. Please see similar comment from Referee n.1.*

Ln 395-400 Specify if you are using the nominal exploration depth of EMI

*REPLY: These aspects concerning the use of apparent electrical conductivity (ECa) maps were already described in section "2.2 available soil maps" where the process behind the creation of the geophysics-based soil map is provided. In the specific lines mentioned by the referee however, the depths are obtained with actual soil coring and we do not think further reference to the geophysical methodology used in this study is needed at this point of the manuscript, as it is extensively described in the previous sections as well as in the publication from Brogi et al. (2018).*

Thanks for the very interesting contribution it was a pleasure to read

*REPLY: Thank you for the nice review, which we think improved the manuscript considerably.*

**Short Comment n.1 (Eric Bönecke)**

This paper "Added value of geophysics-based soil mapping in agro-ecosystem simulations" shows interstitium improvements in combining geophysically based soil mapping with agro-ecosystems modelling. It is well written and suits with the scope of the SOIL

Journal. We agree with the previous comments of the first reviewer and only a have one more specific comment:

L60: refers to the work "Determining the within-field yield variability..." using EMI based soil maps. Actually, the mobile measurement system in this work uses electrical resistivity mapping/tomography (ERT), which is a specific type of measurement using direct current measurements rather than magnetic induced ones. We suggest to rephrase the sentence as "Some case studies successfully used EMI and ERT-based soil maps...". You may also leave out both terms 'EMI' and 'ERT' and merely refer to "geophysically based soil maps".

*REPLY: We thank you for the comment and for the appreciation of our work. Indeed a good point is raised, and we addressed it in this revised version. The new version of this paragraph reads: "Some case studies where geophysics-based soil maps information were successfully combined with crop growth models are available (Boenecke et al., 2018; Wong and Asseng, 2006)".*

―――――――――――――――――――